# BENCHMARKING XAI EXPLANATIONS WITH HUMAN-ALIGNED EVALUATIONS

## ABSTRACT

We introduce PASTA (Perceptual Assessment System for explanaTion of Artificial Intelligence), a novel framework for a human-centric evaluation of eXplainable AI (XAI) techniques in computer vision. Our first key contribution is a human evaluation of XAI explanations on four diverse datasets—COCO, Pascal Parts, Cats Dogs Cars, and MonumAI—which constitutes the first large-scale benchmark dataset for XAI, with annotations at both the image and concept levels. This dataset allows for robust evaluation and comparison across various XAI methods. Our second major contribution is a data-based metric for assessing the interpretability of explanations. It mimics human preferences, based on a database of human evaluations of explanations in the PASTA-dataset. With its dataset and metric, the PASTA framework provides consistent and reliable comparisons between XAI techniques, in a way that is scalable but still aligned with human evaluations. Additionally, our benchmark allows for comparisons between explanations across different modalities, an aspect previously unaddressed. Our findings indicate that humans tend to prefer saliency maps over other explanation types. Moreover, we provide evidence that human assessments show a low correlation with existing XAI metrics that are numerically simulated by probing the model.

## 1 INTRODUCTION

As Deep Neural Networks (DNNs) are used in increasingly high stakes domains (e.g., legal, medical) (Surden, 2021; Litjens et al., 2017), it is essential for humans to interpret how they reach their conclusions (Bender et al., 2021). Their lack of transparency has led them to be characterized as "black boxes" (Castelvecchi, 2016), which is particularly problematic in critical applications where understanding the decision-making process is essential for trust and accountability (Vereschak et al., 2024), leading to the creation of a relatively new field: explainable AI (XAI) (Gunning et al., 2019). XAI aims to make the workings of DNNs more transparent and interpretable. XAI methods fall into two main categories: post-hoc techniques (Selvaraju et al., 2017; Ribeiro et al., 2016; Lundberg & Lee, 2017) and ante-hoc techniques (Bennetot et al., 2022; Koh et al., 2020). Post-hoc techniques generally explain the output of a frozen, pretrained DNN, while ante-hoc techniques modify the architecture of the DNN to improve its interpretability from the outset. Each of these categories can be further subdivided into various sub-families, offering a wide array of XAI approaches.

The diversity of XAI techniques calls for an effort to standardize their evaluation and comparison. Although there are toolkits in computer vision that offer a range of computational evaluation techniques (Agarwal et al., 2022b; Hedström et al., 2023; Fel et al., 2022a), to our knowledge there has been no effort to standardize their evaluation from a perceptual point of view (Nauta et al., 2023), *i.e.,* the way the explanation is perceived by the human for whom it was intended. Currently, prevalent approaches (Dawoud et al., 2023; Colin et al., 2022) to evaluate XAI techniques involve human annotators assessing and ranking their interpretability. This approach aligns with XAI's goal of improving the human interpretability of DNN models. Yet, this method is costly since it requires paying annotators and is impractical for widespread use, as each new XAI technique necessitates a fresh round of human evaluation. It is also at risk of being inconsistent and unreliable since evaluations may differ from one annotator to another and depend on factors such as fatigue and even the time of day (Schmidt et al., 2007).

To address the challenges associated with evaluating XAI techniques, we propose the Perceptual Assessment System for explanaTion of Artificial intelligence (PASTA). PASTA aims to automate the evaluation of XAI techniques by providing an evaluation metric that mimics human preferences.

The first component of PASTA is a dataset composed of four diverse datasets (COCO, Pascal Part, Cats Dogs Cars, and Monumai), which include both image and concept annotations. Using this dataset, we compare 21 XAI methods across multiple model architectures. We subject the resulting explanations to a rigorous evaluation by human annotators, along a comprehensive set of criteria that cover a variety of desired properties.

The second component of PASTA is a metric designed to replicate human evaluation on the PASTA-dataset. While there are benchmarks that focus on perceptual evaluation of XAI methods (Colin et al., 2022; Dawoud et al., 2023), to the best of our knowledge, we are the first to integrate both saliency-based and concept-based explanations into a unified framework. Additionally, our approach addresses multiple dimensions of human assessment by incorporating a diverse set of questions for users. The primary contributions of this paper are as follows:

- **Comprehensive XAI Benchmark**: We establish a dataset, the PASTA-dataset, designed to evaluate XAI methods across various modalities, including vision and concept-based explanations (Sec. 3.1).
- **Extensive Evaluation of XAI Methods**: We conduct a large-scale evaluation of 21 XAI methods, comparing both post-hoc and ante-hoc approaches across multiple datasets (Sec. 3.2—3.4). Our findings indicate that saliency and input perturbation-based techniques, such as LIME and SHAP, are favored for their effectiveness in interpreting model predictions (Sec. 3.5).
- **Human-AI Correlation**: Our findings reveal a low correlation between widely used XAI metrics and human assessments, suggesting that these metrics cover complementary aspects of XAI quality (Sect 3.6).
- **Human-aligned Perception Metric for Explanations**: We introduce a novel, data-based metric, which we call the PASTA-metric, that automates the scoring of XAI techniques along human-like interpretability criteria (Sec. 4).

Automated yet human-aligned metrics such as the PASTA-metric may serve not only to streamline the evaluation process of XAI techniques but also to foster a more transparent and trustworthy AI ecosystem, where DNNs are comprehensible and their decisions justifiable. The complete PASTA framework (code, annotation, and models) will be released after the reviews.

## 2 RELATED WORK

**Automated scoring.** Automated scoring involves developing models that assign scores to inputs based on a reference dataset, often derived from human ratings. A particularly active area of research in this domain is automated essay scoring. Traditionally, this has been addressed through handcrafted feature extraction (Yannakoudakis et al., 2011), but modern methods tend to be closer to model as a judge (Lee et al., 2024; Taghipour & Ng, 2016; Chiang et al., 2024). More recently, there has been a growing interest in using embeddings from large language models (LLMs) as features for scoring. The first successful attempt in this direction was made by Yang et al. (2020). Building on this trend, other approaches have incorporated LLM embeddings with models like LSTMs (Wang et al., 2022), integrated text generation into the training loop (Xiao et al., 2024), or introduced multi-scale aspects to enhance performance (Li et al., 2023).

**Explainable AI.** To address the challenge of explaining DNNs, several specialized tools have been proposed, often categorized into *post-hoc* and *ante-hoc* methods (Arrieta et al., 2020; Rudin et al., 2022). *Post-hoc* methods encompass any tool external to the model, allowing us to gain insights from any pre-trained DNN. Popular examples are GradCAM (Selvaraju et al., 2017), LIME (Ribeiro et al., 2016), and SHAP (Lundberg & Lee, 2017). While most post-hoc explainers agree in providing input regions most responsible for a certain prediction, they differ in many non-trivial details, and selecting and evaluating the most appropriate explainer for each task can be challenging (Leavitt & Morcos, 2020; Roy et al., 2022). *Ante-hoc* methods, instead, aim at modifying the underlying model architecture so as to provide explanations by design. This can be done in the framework of Concept Bottleneck Models (CBMs) (Koh et al., 2020) by prompting the model to first predict a set

Figure 1: **Overview of the human evaluation of the PASTA-dataset.** Each image is paired with multiple computed explanations, which are then annotated by human evaluators, first with baseline questions Q0.1 and Q0.2 and then regarding their interpretability and usefulness, using the questions Q1—Q6 outlined in Section 3.4.

of human-understandable high-level concepts, and then making the final prediction using a shallow and interpretable classifier that supports human inspection, or by decomposing the *reasoning* of the model into smaller and more actionable steps (Ge et al., 2023).

**Evaluating explainability.** While several methods have been proposed to quantitatively measure explanation quality, such as faithfulness (Petsiuk et al., 2018; Dasgupta et al., 2022), sparsity (Chalasani et al., 2020; Bénard et al., 2021), robustness (Alvarez-Melis & Jaakkola, 2018b; Montavon et al., 2018), sensitivity (Adebayo et al., 2018; Hedström et al.) and alignment to an assumed ground truth (Colin et al., 2022; Mohseni et al., 2021; Dawoud et al., 2023), they inherently overlook the perceptual aspect with respect to the human, which is the expected consumer of such explanations. Evaluating explanations via user studies, e.g. where annotators are asked to rate and evaluate explanations (Chen et al., 2018; Shu et al., 2019), are however very costly, prone to unreproducibility issues (Nauta et al., 2023), and often unfeasible for tasks that require trained users, like in the medical domain (Miró-Nicolau et al., 2022; Muddamsetty et al., 2021). In this work, we take the first step towards standardizing the evaluation of human perception preferences of explanations (Nauta et al., 2023). We propose to overcome the issues of hard-to-reproduce large-scale user studies by automatizing the evaluation of XAI techniques through a multi-value scoring system that mimics human preferences while taking into account the users' diverse expectations, which naturally emerge in user-based studies.

## 3 CREATING A HUMAN PREFERENCE DATASET FOR XAI INTERPRETABILITY

To evaluate the quality of XAI explanations from a human-centric point of view, we proceed along the following steps, which are detailed in the subsections below. First, we build a dataset from annotated images. Using our codebase of classifiers and XAI techniques, we compute label predictions along with XAI explanations. The explanations are then evaluated by human annotators following a rigorous evaluation protocol. Finally, we compare the different XAI techniques with the human evaluations to assess quality of their explanations. We also investigate how human scores correlate with popular automated XAI metrics to see whether they are complementary.

### 3.1 DATASET COMPOSITION

To evaluate the performance of different perceptual metrics, we collect a large-scale dataset composed of images of four highly heterogeneous datasets: COCO (Lin et al., 2014), Pascal Part (Chen et al., 2014), Cats Dogs Cars (Kazmierczak et al., 2024), and Monumai (Lamas et al., 2021). This

collection is referred to as the classifier's dataset. The precise set of labels used for each classifier's dataset is described in Section A.1.1.. It is important to distinguish this from the PASTA-dataset, which comprises the final set of images, annotations, labels, and explanations obtained through our evaluation process. Each dataset includes two levels of annotations: image-based annotations and concept-based annotations. This dual-level annotation framework allows for the application of Concept-Based and other XAI methods, enabling a robust evaluation across different approaches. For this, we use for the computation of explanations a subset of 25 images of the classifier's dataset test split per dataset, resulting in a comprehensive evaluation of 100 images, that serve as the basis of the PASTA-dataset. This diverse selection ensures a broader generalization of the XAI techniques across datasets being assessed. Note that, unlike traditional datasets, our benchmark dataset comprises a triplet of images, explanations, and labels. This triplet enables us to quantitatively assess the quality of XAI techniques (see Figure 1).

## 3.2 XAI Methods

To ensure representativity, we consider two distinct types of explanations. The first type comprises saliency methods, which generate explanations by assigning an importance score to each pixel of the input image, indicating the significance of each pixel in the prediction process. The second type consists of concept-based explanations, which highlight the importance of human-understandable concepts in the explanation. A detailed list of the methods and more details on each technique are given in Appendix A.2. Notably, we present in this appendix a succinct definition of each XAI method used.

Among saliency-based methods, we consider model-agnostic explanations (LIME (Ribeiro et al., 2016) and SHAP (Lundberg & Lee, 2017)), gradient-based (FullGrad (Srinivas & Fleuret, 2019)) and model-specific techniques. The model-specific methods include GradCAM (Selvaraju et al., 2017), HiResCAM (Draelos & Carin, 2020), GradCAMElementWise (Pillai & Pirsiavash, 2021), GradCAM++ (Chattopadhay et al., 2018), XGradCAM (Fu et al., 2020), AblationCAM (Ramaswamy et al., 2020), ScoreCAM (Wang et al., 2020), EigenCAM (Muhammad & Yeasin, 2020), EigenGradCAM (Muhammad & Yeasin, 2020), LayerCAM (Jiang et al., 2021), Deep Feature Factorizations (Collins et al., 2018), and BCos (Böhle et al., 2024).

Among concept-based methods, we explore those that produce explanations through counterfactuals (CLIP-QDA-sample (Kazmierczak et al., 2024)), and feature importance (X-NeSyL (Díaz-Rodríguez et al., 2022), LaBo (Yang et al., 2023), CLIP-linear (Yan et al., 2023), LIME-CBM (Kazmierczak et al., 2024), RISE (Petsiuk et al., 2018) and SHAP-CBM (Kazmierczak et al., 2024)). Additionally, we employ various strategies for concept extraction: zero-shot methods, training from concept annotations, and training from bounding boxes.

A beneficial aspect of our approach is that many of the methods are tested on multiple backbone architectures (see Figure 7). In such cases, the XAI method is applied to independently trained models. For instance, GradCAM is evaluated on ResNet50, ViT-B, and CLIP-zero-shot models, while SHAP-CBM is applied to both CLIP-QDA and the original Concept Bottleneck model as proposed by Koh et al. (2020).

## 3.3 Training Classifiers and Computing the XAI Dataset

The initial phase in constructing the PASTA-dataset involves training the various classifier models on which explanations will be generated. Specifically, we utilize ResNet50 (He et al., 2016), ViT-B (Dosovitskiy, 2020), ResNet50-BCos (Böhle et al., 2024), CLIP-Linear (Yan et al., 2023), CLIP-QDA (Kazmierczak et al., 2024), X-NeSyL (Díaz-Rodríguez et al., 2022), and ConceptBottleneck (Koh et al., 2020). These models are trained separately on the each classifier's dataset evoqued in Section 3.1. The final assessment of XAI techniques is conducted on samples of the test set. To provide a diverse range of explanations, our framework incorporates both black-box models, explained using post-hoc methods, and ante-hoc methods, specifically CBMs. The specific details regarding these two families of methods like the set of concepts used in CBMs are presented in Appendix A.1.2.

Our codebase includes the 21 XAI methods described in Sec. 3.2 and the 7 backbone models described above. Some XAI methods are incompatible with certain backbones, see Table 7, this leaves

Figure 2: **Samples corresponding to questions 5 and 6.** On the left, a light perturbation is applied, resulting in no label change. On the right, a strong perturbation is applied, leading to a label change.

46 distinct combinations of XAI methods and backbones, which we refer to as *XAI techniques*. We apply each technique to 100 images. This leads to an XAI dataset of 4600 instances, each of which is associated with an image and its ground truth label, the label prediction from a classifier instance, and the explanation from a particular XAI technique. In the next section, we present an approach for evaluating the human perception of these instances.

## 3.4 HUMAN EVALUATION PROTOCOL

We aim to quantify the interpretability and usefulness of XAI techniques accurately, using a human evaluation of the quality of evaluations. The resulting dataset serves as a benchmark, enabling us to compare and validate current and future XAI methods. Our human-centric approach complements existing approaches that focus primarily on assessing the model's internal behavior. For example, traditional evaluations of *faithfulness* measure how closely an explanation corresponds to the model's true functioning while we assess in our dataset how the explanation fit human expectations.

We take a structured approach to ensure that the explanations are not only technically sound but also align with human cognitive processes and expectations, fostering the development of more transparent and interpretable AI systems. First, we establish a comprehensive set of assessment criteria that are evaluated on a graded scale. Then, we apply a meticulous evaluation protocol, developed with the help of a psychologist, to ensure that annotators fully understand the task and the expectations. This includes annotator training and close monitoring throughout the process.

**Evaluation Criteria** We consolidate different criteria from the literature into the following set of *desiderata* for XAI explanations that we wish to evaluate (equations, definitions and algorithms defined in Appendix B.2):

- *Faithfulness* (Arrieta et al., 2020; Fel et al., 2022b) measures the extent to which an explanation accurately reflects the true behaviour of the model.
- *Robustness* (Doshi-Velez & Kim, 2017; Agarwal et al., 2022a; Yeh et al., 2019) assesses the stability and relevance of the explanation across a broad range of models and inputs.
- *Complexity* (Nauta et al., 2023; Nguyen & Martínez, 2020; Bhatt et al., 2021) checks whether the explanation is both simple and informative, balancing clarity and detail.
- *Objectivity* (Bennetot et al., 2022) evaluates whether the explanation is interpreted consistently by the majority within a given audience.

**Evaluation Protocol** Gathering human preferences from surveys is an active field of research in psychology (Fowler Jr, 2013), but also in the field of machine learning *e.g.,* with the recent advent of reinforcement learning from human feedback (RLHF) (Kaufmann et al., 2023). Interfaces, as well as the formulation of questions, play a key role in the quality of the annotations (Pommeranz et al., 2012), and their design must be considered cautiously to avoid confounding cognitive biases. The following human evaluation protocol has been designed with the help of a psychologist. The

formulation of the questions has been carefully chosen to ensure that they are fully understood by each annotator.

To maintain consistency and reliability, all annotators undergo a training session before starting the actual annotation task. This training familiarizes them with the XAI techniques, evaluation criteria, rating scale, and datasets, ensuring a uniform understanding of the task and the expectations. More details about the annotation protocol are given in Appendix A.3, including an example of how questions are presented to the annotators (Fig. 9).

During the evaluation process, annotators are shown an image, a prediction, and an explanation. They are then asked a list of questions that we now describe in more detail. A first set of questions aims at having annotators establish a baseline, i.e., by interpreting and explaining what makes an image recognizable as a specific object or class. This prompts the human annotator to think about what they are relying on to classify the image themselves, before having to evaluate explanations produced by the XAI techniques. The first two questions in the annotation process are:

- Q0.1: What part makes you classify this image as ***? (write an explanation extracting concepts)
- Q0.2: What part of the input helps the prediction? (draw bounding boxes on the image)

Each explanation generated by the XAI techniques is then evaluated regarding each of the desiderata given above (indicated in italics):

- Q1: The provided explanation is consistent with how I would explain the predicted class? *Fidelity*
- Q2: Overall the explanation provided for the model prediction can be trusted? *Complexity*
- Q3: Is the explanation easy to understand? *Complexity*
- Q4: The explanation can be understood by a large number of people, independently of their demographics (age, gender, country, etc.) and culture? *Objectivity*
- Q5: With this perturbed image, to what extent has the explanation changed ? (Examples with good predictions and light perturbations) *Robustness*
- Q6: With this perturbed image, to what extent has the explanation changed? (Examples with bad predictions and strong perturbations) *Robustness*

Annotators evaluate how well the explanations conform to their expectations (Q1), whether the explanations are clear (Q2 and Q3), and if they would rely on these explanations (Q4). They also assess how much the explanations change when the images are perturbed (Q5 and Q6), for both accurate and inaccurate predictions, as studied by (Fel et al., 2022b). An example is shown in Figure 2. Each explanation is rated on a scale from one to five stars, where one star indicates the explanation is entirely uncorrelated with the annotator's reasoning, and five stars represent perfect correlation. This star rating system allows for a nuanced assessment of the quality of the explanations, reflecting how closely they align with human understanding.

## 3.5 HUMAN EVALUATION AND RESULTS

We now present a brief summary of the human evaluations obtained in the PASTA-dataset, using the previously described protocol. Full results and values are available in Appendices B.3 and B.4. The PASTA-dataset, described in Sec. 3.3, contains 4600 instances with images, predictions, and explanations. From this set, we select 2200 samples randomly and let them be evaluated by humans according to the protocol above. Each instance receives five evaluations from different annotators. We aggregate these evaluations using majority voting to favor consensus opinions. As illustrated by Figure 3a, we observe that these results indicate a preference for image-based techniques, suggesting that saliency maps are perceived as more interpretable than CBMs. There are several potential reasons why saliency-based methods might be preferred over concept-based ones. First, this could be attributed to the more active research field surrounding saliency methods, as discussed below. Another plausible reason is the straightforwardness of the explanation provided by saliency maps, which is especially relevant given that our dataset is oriented toward non-expert annotators. Additionally, Figure 3b shows the average score among techniques that share the same backbone. Interestingly, CLIP and ViT have similar scores, likely due to the architectural similarities between the two models. ResNet 50, which played a pivotal role in the development of many XAI methods, consistently scores higher. This could suggest a potential bias toward ResNet 50 in the design and

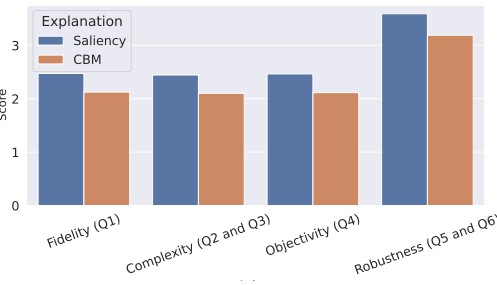 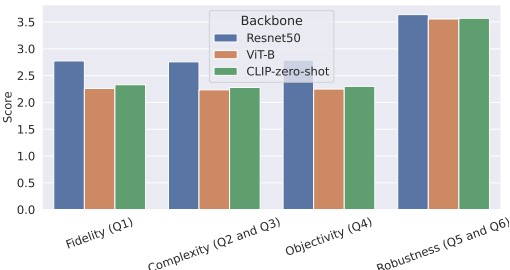

(a) Scores for each question, for saliency-based and CBM-based explanation

(b) Scores for each question, for different backbones of saliency methods

Figure 3: **Comparison of explanation methods and backbones:** Overall, saliency methods are preferred over CBM-based explanations. As backbones for saliency methods, ViT-B and CLIP obtain overall similar results, while Resnet50 has better scores.

Table 1: **Pearson Correlation Coefficient (PCC) and Spearman rank Correlation Coefficient (SCC) between** *faithfulness* **computed with different perturbation strategies and human scores.** In parentheses, the respective p-values.

|  | ROAD | | Black patches | | Uniform noise | | Gaussian noise | |
|---|---|---|---|---|---|---|---|---|
|  | PCC | SCC | PCC | SCC | PCC | SCC | PCC | SCC |
| **Q1** | 0.01 (0.62) | -0.04 (0.12) | 0.06 (0.02) | 0.05 (0.04) | 0.07 (0.01) | 0.03 (0.24) | 0.07 (0.02) | 0.02 (0.38) |
| **Q2** | -0.01 (0.88) | -0.03 (0.20) | 0.04 (0.19) | 0.04 (0.17) | 0.06 (0.04) | 0.02 (0.45) | 0.05 (0.08) | 0.01 ( 0.63) |
| **Q3** | 0.03 (0.33) | -0.03 (0.34) | 0.04 (0.10) | 0.04 (0.16) | 0.08 (0.01) | 0.04 (0.13) | 0.06 (0.02) | 0.03 (0.30) |
| **Q4** | 0.03 (0.25) | -0.04 (0.17) | 0.04 (0.19) | 0.03 (0.33) | 0.08 (0.01) | 0.03 (0.21) | 0.06 (0.02) | 0.02 (0.41) |
| **Q5** | -0.05 (0.05) | 0.02 (0.41) | 0.01 (0.93) | 0.04 (0.14) | -0.05 (0.07) | -0.01 (0.78) | -0.04 (0.16) | 0.01 (0.95) |
| **Q6** | -0.02 (0.37) | 0.06 (0.03) | -0.13 (1e-5) | -0.08 (0.01) | -0.04 (0.16) | -0.01 (0.68) | -0.04 (0.13) | -0.01 (0.79) |

effectiveness of current XAI methods. Another method that seems to perform well is EigenCAM. Notably, the method does not rely on class discrimination results, which simplifies the process of showcasing salient objects. This often leads to the generation of saliency maps that are both plausible and easy to interpret. However, these maps tend to be less faithful to the model's actual decision-making process. This discrepancy underscores the distinction between human agreement—what users perceive as important—and the model's faithfulness.

### 3.6 CORRELATION WITH OTHER METRICS

We turn to the question of how the human scores in the PASTA-dataset correlate with standard XAI metrics. An analysis based on the Pearson Correlation Coefficient and the Spearman rank Correlation Coefficient for different perturbation strategies, shown in Table 1 indicates a rather weak correlation between human scores and ROAD (Rong et al., 2022), a popular metric to evaluate faithfulness. We conclude that our human scores indeed cover an aspect of explanation quality unrelated to that of perceptual quality, as predicted by Biessmann & Refiano (2021). Additional results, including results for other axioms, are available in Appendix B.2 and B.3.

Then, our findings reinforce the idea that human evaluations and computational metrics measure complementary aspects of XAI methods. Human evaluations excel at assessing the usefulness of explanations, aligning with their primary purpose of serving a human audience. In contrast, computational metrics, such as faithfulness, focus on evaluating the alignment between the explanation and the model's actual internal functioning. This aspect lies beyond the reach of human judgment, as humans cannot directly access or fully comprehend the internal mechanisms of the model.

## 4 DEVELOPING A METRIC FOR PERCEPTUAL EVALUATION

Our human preference dataset contains evaluations of the fidelity, complexity, objectivity, and robustness of each evaluation. These scores were painstakingly attributed by human annotators. To provide a tool for measuring human assessment of XAI techniques, we introduce the PASTA-metric,

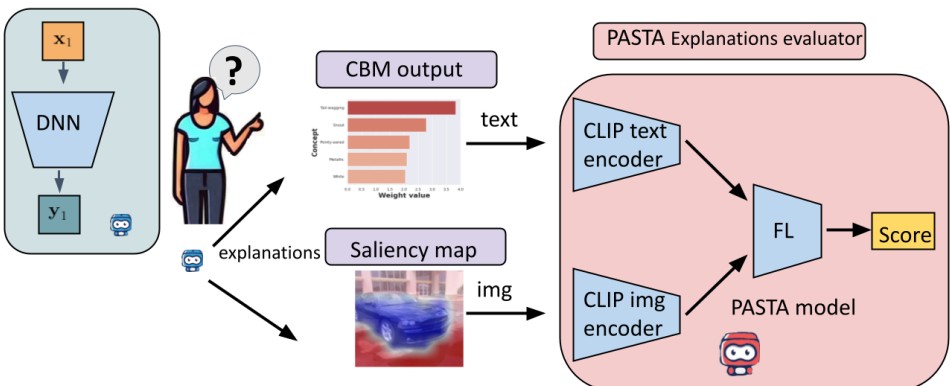

Figure 4: **Pipeline of the PASTA-metric.** Initially, embeddings are computed based on the explanations. Then a scoring network, trained on labels provided by the Benchmark Dataset, generates a final score.

which simulates human evaluation. The global pipeline is illustrated in Figure 4. More precisely, the PASTA-metric is composed of an embedding network (Section 4.1), that processes both CBM outputs or saliency maps, and a scoring network (Section 4.2), that computes scores from the embeddings. Using the data collected in Section 3, the PASTA-metric aims at predicting the human scores for questions Q1 to Q6.

## 4.1 COMPUTATION OF EMBEDDINGS

Drawing inspiration from recent literature in automated scoring (Yang et al., 2020; Wang et al., 2022), we use a foundation model to generate embeddings. Given its multimodal capabilities, we select CLIP (Yan et al., 2023) as the embedding model. This choice allows for a unified integration of both concept-based explanations, which can be transformed into text, and saliency map-based explanations, which can be projected into the same embedding space.

Let us denote by $x_i \in \mathbb{R}^{H \times W \times 3}$ the $i$-th test image of height $H$ and width $W$ in the dataset and by $e_i^{saliency} \in \mathbb{R}^{H \times W}$ any explanation produced by a saliency-based XAI method for this image. We also note the CLIP image encoder as $\text{CLIP}_{\text{image}}$. For saliency explanations, the resulting embedding based on a saliency map can be obtained using the following formula:

$$\phi_{\text{image}}(e_i^{saliency}) = \text{CLIP}_{\text{image}}(Heatmap(x_i, e_i^{saliency})), \tag{1}$$

where $Heatmap$ is the process generating the saliency related heatmap to the image.

For CBMs, let us denote with $e_i^{CBM} \in \mathbb{R}^K$ any explanation produced by a saliency-based XAI method for this image $x_i$, where $K$ is the length of the concept set. These attributions are converted into a sentence, which is then embedded. Let $\text{CLIP}_{\text{text}}$ denote the CLIP text encoder and $Sentence$ the process of converting the CBM explanation into text. The resulting embedding is:

$$\phi_{\text{text}}(e_i^{CBM}) = \text{CLIP}_{\text{text}}(Sentence(e_i^{CBM})). \tag{2}$$

By default, the described process is applied; however, we also offer alternatives using LLaVa (Liu et al., 2024). Another variant, with handcrafted features as multimodal encoders detailed in Section D is also tested. Plus, we offer additional ablation about alternative ways to process Equations 1 and 2 in Appendix D, like the influence of $K$ or the use of templates in textual descriptions.

## 4.2 SCORING NETWORK

Once the embeddings are computed, a scoring network composed of a linear layer is used to predict scores. Inspired by Automated Essay Scoring (Yang et al., 2020; Wang et al., 2022), we use a loss $L$ that combines a similarity loss $L_s$, a mean squared error (MSE) loss $L_{mse}$, and a ranking loss $L_r$. From a set of ground truth scores obtained from majority voting $\{m_k\}_{k \in [0, N_s]}$ and the predictions

Table 2: **Training and Inference Times on GPU and CPU.**

| Device | Training Time (s) | Inference Time (s) |
|--------|-------------------|--------------------|
| GPU    | 214.18            | 8.89               |
| CPU    | 1308.84           | 57.52              |

given by the scoring network $\{\hat{m}_k\}_{k\in[0,N_s]}$, the resulting loss is defined as:

$$L = \alpha L_s + \beta L_{mse} + \gamma L_r, \tag{3}$$

where $\alpha$, $\beta$, and $\gamma$ are hyperparameters controlling the relative importance of each component. Formulas about the different losses are given in Appendix C.5 The PASTA-dataset presenting 5 ground truth votes for a given inference, a discussion have been made about how to aggregate the votes. After ablation (Done in Appendix C.2), we choose to use mode (formula given in Appendix C.1.1).

### 4.3 CLASSIFIER RESULTS

In all experiments, we employed the Adam optimizer (Kingma & Ba, 2017) with a batch size of 128, training for 500 epochs at a learning rate of 0.001. Additionally, we configured the parameters as follows: $\alpha$ was set to 1, $\beta$ to 0.01, and $\gamma$ to 0.1. We split our dataset in 1540 training samples, 330 validation samples, and 330 testing samples. The experiments were conducted using a V100-16GB GPU. The training and inference times are summarized in Table 2. It is important to note that, for both inference and testing, the majority of the computational time is dedicated to precomputing CLIP embeddings, as the scoring network itself is relatively lightweight and requires minimal computational resources.

For questions 1 to 6 in Section 3, we calculated the Mean Square Error (MSE), Quadratic Weighted Kappa (QWK), and Spearman Correlation Coefficient (SCC) between the predicted and ground truth labels on the test set (formulas given in Appendix C.1.2). The results are presented in Table 3. For comparison, we also include the scores obtained using CLIP and LLaVA (Liu et al., 2024) as a multimodal encoder, denoted as PASTA-metric[CLIP] and PASTA-metric[LLaVa]. We also tested an alternative using handcrafted features, with a process described in Section D, denoted by Feature Extraction. Finally, we report the inter-annotator agreement values, which correspond to the metrics computed between a randomly selected annotator's score and the mode.

We first observe that PASTA-metric[CLIP] and PASTA-metric[LLaVa] yield similar results. Therefore, we allow users to choose between these variants based on their specific needs. PASTA-metric[CLIP] offers the shortest training and inference times (see Table 2 ), but users should be aware that CLIP is known to suffer from bias, as highlighted by Moayeri et al. (2023). On the other hand, PASTA-metric[LLaVa] requires more computational time but is less prone to biases associated with contrastive pre-training. Finally, the approach using handcrafted feature extraction guarantees a fully interpretable process, though it produces less satisfactory results.

### 4.4 GENERALIZATION CAPABILITIES OF THE PASTA-METRIC

In the main study, to constitute training, validation, and test sets, we shuffled all the samples considering the image they belong to. In this section, we investigate the impact of shuffling instead. By doing so, we ensure that samples from the same XAI technique cannot be in two different splits. This will help us investigate the generalization capabilities of the model in two distinct ways: can it generalize to new XAI techniques? The results of the two setups for Q1 are shown in Table 4. We also, consider the variant that shuffles all the samples without considering the XAI technique or the dataset they belong to.

The results indicate a decrease of 0.04 in QWK when shuffling across XAI techniques and a more significant drop of 0.07 when shuffling across image IDs. This opens a discussion on the potential for applying the PASTA-metric to other image datasets. Regarding the generalization to new XAI methods, the relatively moderate drop in performance supports the feasibility of testing our metric on novel XAI techniques.

Table 3: **Mean Square Error (MSE), Quadratic Weighted Kappa (QWK), and Spearman Correlation Coefficient (SCC) for each question.** Each value is the average of 5 runs with standard deviation. *Human* refers to inter-annotator agreement.

| Metric | Model | Q1 | Q2 | Q3 | Q4 | Q5 | Q6 |
|---|---|---|---|---|---|---|---|
| MSE | PASTA-metric$^{\text{CLIP}}$ | $1.06 \pm 0.05$ | $1.13 \pm 0.09$ | $\mathbf{1.21 \pm 0.13}$ | $1.15 \pm 0.13$ | $1.96 \pm 0.27$ | $\mathbf{0.76 \pm 0.21}$ |
| | PASTA-metric$^{\text{LLaVa}}$ | $\mathbf{1.02 \pm 0.20}$ | $\mathbf{1.04 \pm 0.24}$ | $1.28 \pm 0.34$ | $\mathbf{1.08 \pm 0.28}$ | $\mathbf{1.66 \pm 0.23}$ | $1.13 \pm 0.11$ |
| | Feature Extraction | $4.50 \pm 0.40$ | $5.81 \pm 0.46$ | $3.71 \pm 0.74$ | $3.61 \pm 0.35$ | $3.54 \pm 0.33$ | $3.58 \pm 0.41$ |
| | *Human* | $0.53 \pm 0.03$ | $0.51 \pm 0.05$ | $0.74 \pm 0.03$ | $0.72 \pm 0.02$ | $1.00 \pm 0.06$ | $0.52 \pm 0.03$ |
| QWK | PASTA-metric$^{\text{CLIP}}$ | $\mathbf{0.48 \pm 0.05}$ | $0.44 \pm 0.08$ | $\mathbf{0.43 \pm 0.07}$ | $\mathbf{0.43 \pm 0.07}$ | $0.32 \pm 0.07$ | $\mathbf{0.48 \pm 0.13}$ |
| | PASTA-metric$^{\text{LLaVa}}$ | $0.43 \pm 0.12$ | $\mathbf{0.45 \pm 0.08}$ | $0.40 \pm 0.11$ | $0.42 \pm 0.10$ | $\mathbf{0.36 \pm 0.09}$ | $0.00 \pm 0.00$ |
| | Feature Extraction | $0.09 \pm 0.04$ | $0.09 \pm 0.02$ | $0.03 \pm 0.04$ | $0.03 \pm 0.03$ | $0.05 \pm 0.02$ | $0.02 \pm 0.02$ |
| | *Human* | $0.73 \pm 0.03$ | $0.74 \pm 0.02$ | $0.63 \pm 0.02$ | $0.62 \pm 0.02$ | $0.65 \pm 0.03$ | $0.59 \pm 0.02$ |
| SCC | PASTA-metric$^{\text{CLIP}}$ | $\mathbf{0.25 \pm 0.25}$ | $0.23 \pm 0.24$ | $\mathbf{0.23 \pm 0.23}$ | $0.22 \pm 0.23$ | $0.17 \pm 0.17$ | $\mathbf{0.24 \pm 0.25}$ |
| | PASTA-metric$^{\text{LLaVa}}$ | $0.23 \pm 0.24$ | $\mathbf{0.24 \pm 0.25}$ | $0.21 \pm 0.23$ | $\mathbf{0.22 \pm 0.24}$ | $\mathbf{0.20 \pm 0.20}$ | $0.00 \pm 0.00$ |
| | Feature Extraction | $0.16 \pm 0.21$ | $0.09 \pm 0.09$ | $0.14 \pm 0.30$ | $\mathbf{0.22 \pm 0.32}$ | $0.17 \pm 0.17$ | $0.11 \pm 0.11$ |
| | *Human* | $0.37 \pm 0.37$ | $0.38 \pm 0.38$ | $0.33 \pm 0.33$ | $0.33 \pm 0.33$ | $0.34 \pm 0.34$ | $0.29 \pm 0.29$ |

Table 4: **Summary of results for different restrictions applied during dataset splitting.** The label *No* indicates no restrictions, *Img id* denotes that the same image indices $i$ are maintained across splits, and *XAI id* indicates that the same explanation indices $j$ are preserved in different splits. Each value is the average result on 5 runs with the standard deviation.

| Restriction split | MSE | QWK | SCC |
|---|---|---|---|
| *No* | $\mathbf{0.85 \pm 0.06}$ | $\mathbf{0.55 \pm 0.04}$ | $\mathbf{0.28 \pm 0.28}$ |
| *xai_id* | $0.96 \pm 0.10$ | $0.51 \pm 0.05$ | $0.26 \pm 0.26$ |
| *img_id* | $1.06 \pm 0.05$ | $0.48 \pm 0.05$ | $0.25 \pm 0.25$ |

## 5 CONCLUSIONS

In this paper, we introduce PASTA, a novel perceptual assessment system designed to benchmark explainable AI (XAI) techniques in a human-centric manner. We integrate four diverse datasets — COCO, Pascal Parts, Cats Dogs Cars, and Monumai — to form a large-scale benchmark dataset for XAI, and used it for an assessment of XAI explanations by human annotators. We also develop an automated evaluation metric that mimics human preferences based on a comprehensive database of human evaluations. This framework offers a scalable and reliable way to compare different XAI methods, facilitating robust evaluations across modalities previously unaddressed.

Our findings demonstrate a clear preference for saliency-based explanations, particularly techniques such as LIME and SHAP, which align well with human intuition. These results affirm the scalability and reliability of our perceptual metric, which provides consistency with human assessment while automating much of the evaluation process.

However, there are limitations to our approach. The current study focuses on a fixed set of datasets and XAI techniques. Human evaluations can be influenced by subjective factors that may affect the consistency of results. Furthermore, annotators can inadvertently introduce biases. Perceptual metrics like the ones proposed here are therefore not intended to serve as absolute measures of XAI performance. Rather, we consider them simply as complementary to other XAI metrics.

Looking ahead, dynamic scoring approaches could be explored to capture the evolving nature of XAI techniques and their use in real-world applications. In conclusion, PASTA intends to take a step towards creating a transparent and trustworthy AI ecosystem. By aligning AI explanations with human cognitive processes, we aim to foster the development of more interpretable AI systems that can be understood and trusted by users across various domains. This work also introduces a perceptual metric, paving the way for future research to implement the PASTA-metric as a perceptual loss aimed at enhancing the trustworthiness of networks, drawing for example inspiration from the emerging use of LPIPS (Zhang et al., 2018) in tasks such as image generation (Jo et al., 2020). Another possible use is XAI method hyperparameter fine-tuning, as proposed in Section E.2.

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

TABLE OF CONTENTS - SUPPLEMENTARY MATERIAL

# A PASTA-DATASET: PROCESS

## A.1 CLASSIFIER TRAINING

### A.1.1 DATASET

The dataset used in PASTA is designed to provide a benchmark for evaluating a wide range of XAI techniques across different explanation modalities. To ensure robustness and versatility, we have built a benchmark dataset consisting of four diverse, publicly available datasets, each bringing distinct characteristics in terms of visual content and concept annotations. Choosing which task to focus on is a tough question. We have chosen to focus on the image classification task. This task can be performed in many different domains, but in order not to be too domain-specific, we decided to work on general datasets. These datasets enable the evaluation of both image-based and concept-based XAI methods.

The dataset used to train our inference model integrates four datasets, each chosen for its unique attributes that align with the requirements of evaluating explainability methods. The datasets are as follows:

- COCO: A widely-used dataset known for its complexity and variety, containing 117k training images, 4.5k validation images annotated with 80 object categories, which we consider to be concepts in the images. The labels correspond for this specific dataset to indoor scene labeling, to do so, we took the subset of images of indoor scenes (53,051 images). Then, we labeled the images using a scene label DNN trained on the MIT SUN.

- Pascal Part: This dataset focuses on detailed part-level annotations, providing fine-grained insights into object structure and component relationships. It is composed of 13,192 training images, 39 concepts, and 16 classes.

- Cats Dogs Cars: A curated dataset featuring images of cats, dogs, and cars. The goal of this dataset is to explore if color biases are present in the model or not. It is composed of 3,858 training images, 39 concepts, and 3 classes. Since this network does not include annotated concepts, we used Grounding DINO (Liu et al., 2023) as an annotator. Since the number of images that constitute Cate Dogs Cars is sufficiently small, we manually checked the bounding boxes generated and found no significant errors.

- Monumai: A specialized dataset containing images of monuments, with annotations that include both the overall structures and specific architectural features. It is composed of 908 images, 15 concepts, and 4 classes.

Each dataset in the classifier's training datasets is annotated at two levels:

- Image-level annotations: These are traditional class labels (Table 6) or object categories that describe the primary content of the image.
- Concept-level annotations: These describe specific, human-understandable features within the image, enabling the application of Concept Bottleneck Models (CBMs) and other concept-based XAI methods. The list of concepts for each dataset is detailed in Table 5.

This dual-level annotation setup ensures that XAI methods can be evaluated not only for their ability to explain class predictions but also for how well they handle concept-based explanations. The presence of both granular (part-level, concept) and holistic (object, scene-level) annotations provides a comprehensive evaluation environment for various XAI methods.

In Figure 5, we observe the class distribution across the different datasets. While the distributions are not perfectly uniform, they generally reflect the original composition of the datasets, ensuring that the diversity of the data is preserved in the evaluation process.

### A.1.2 SPECIFIC PROCEDURES FOR CBMS AND BLACK BOXES MODELS.

To explain the various training procedures for our CBMs, we decompose them into two components: the concept extractor and the classifier. The concept extractor generates an embedding from an input image, with each element representing a concept, while the classifier predicts the label from

Table 5: **List of concepts used in all our CBMs.** For each *Dataset* used, we choose a different set to fit the annotations.

| Dataset | Concepts |
|---|---|
| **catsdogscars, pascalpart** | engine, artifact_wing, animal_wing, stern, tail, locomotive, arm, hair, wheel, chain_wheel, handlebar, hand, headlight, saddle, body, bodywork, beak, head, eye, foot, leg, neck, torso, cap, license_plate, door, mirror, window, ear, muzzle, horn, nose, hoof, mouth, eyebrow, plant, pot, coach, screen |
| **monumai** | horseshoe-arch, lobed-arch, pointed-arch, ogee-arch, trefoil-arch, serliana, solomonic-column, pinnacle-gothic, porthole, broken-pediment, rounded-arch, flat-arch, segmental-pediment, triangular-pediment, lintelled-doorway |
| **coco** | person, backpack, umbrella, handbag, tie, suitcase, bicycle, car, motorcycle, airplane, bus, train, truck, boat, traffic light, fire hydrant, stop sign, parking meter, bench, bird, cat, dog, horse, sheep, cow, elephant, bear, zebra, giraffe, frisbee, skis, snowboard, sports ball, kite, baseball bat, baseball glove, skateboard, surfboard, tennis racket, bottle, wine glass, cup, fork, knife, spoon, bowl, banana, apple, sandwich, orange, broccoli, carrot, hot dog, pizza, donut, cake, chair, couch, potted plant, bed, dining table, toilet, tv, laptop, mouse, remote, keyboard, cell phone, microwave, oven, toaster, sink, refrigerator, book, clock, vase, scissors, teddy bear, hair drier, toothbrush |

Table 6: **List of classes used in all the datasets used to train our inference models.**

| Dataset | Labels |
|---|---|
| **catsdogscars** | cat, dog, car |
| **pascalpart** | aeroplane, bicycle, bird, bottle, bus, car, cat, cow, dog, horse, motorbike, person, pottedplant, sheep, train, tvmonitor |
| **monumai** | Baroque, Gothic, Hispanic-Muslim, Renaissance |
| **coco** | shopping_and_dining, workplace, home_or_hotel, transportation, sports_and_leisure, cultural |

this embedding. We categorize the CBMs we use based on the training methods for these two components. For CLIP-based CBMs (LaBo, CLIP-linear, and CLIP-QDA), the concept extraction is performed in a zero-shot manner *i.e.*, we only use the training images and labels to train the classifier. For CBMs that require training the concept extractor, we use the concept annotations provided by each dataset.

For explanations that involve the application of post-hoc techniques on black-box models, we selected the following DNNs: ResNet 50, ViT, and CLIP (zero-shot). For ResNet 50 and ViT, a separate network was trained for each dataset. For CLIP (zero-shot), we followed the standard procedure proposed by Radford et al. (2021), which classifies by selecting the highest similarity score between the image embedding and all the text embeddings. For post-hoc explanations, we directly extract the explanation after training.

### A.1.3 RESULTS

As illustrated in Figure 6, the models used in this study achieve an accuracy of at least 59%. Notably, one of the models, the zero-shot CLIP, exhibits difficulty specifically with the Monumai dataset, which explains some of the performance variability. Despite this, the overall accuracy of the models remains relatively consistent across datasets. For CBMs, achieving high accuracy across all models required certain compromises, particularly with respect to the concepts used. Although for uniformity we used the same concept sets across different models, it was not always guaranteed that the trained model is the best model.

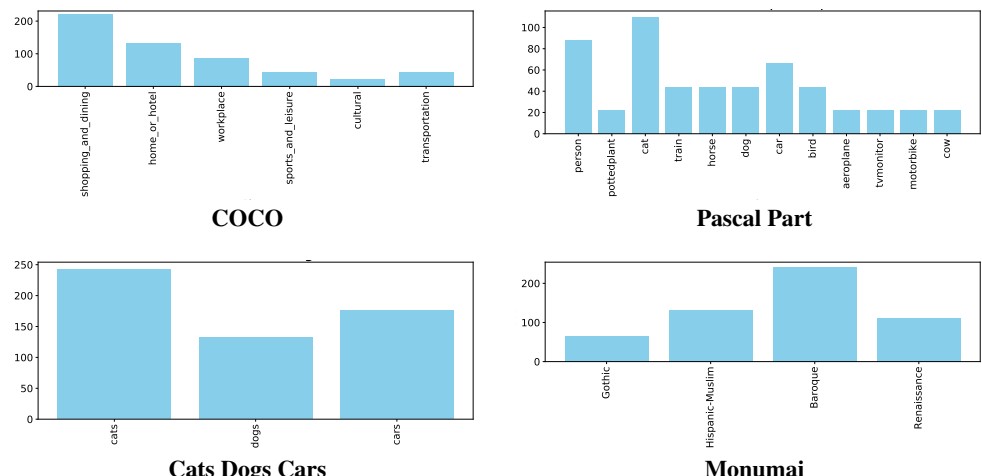

Figure 5: **Class distribution across the different test sets.**

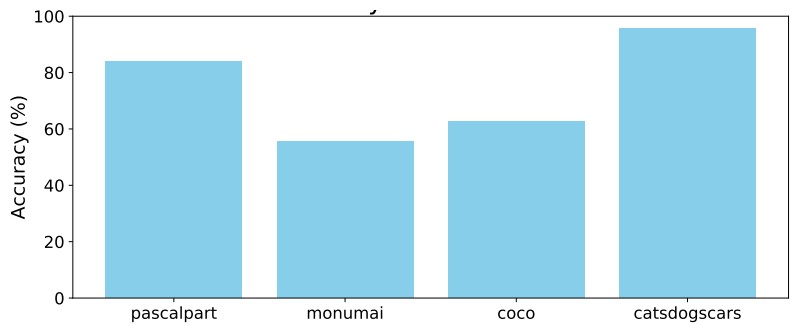

Figure 6: **Accuracy across the different test sets of the different models.**

## A.2 XAI TECHNIQUES

Table 7 presents an overview of the different XAI methods integrated into the dataset. A brief description of each method is provided below to summarize their key features and mechanisms.

**LIME (Local Interpretable Model-agnostic Explanations)**: LIME explains individual predictions of any classifier by approximating it locally with an interpretable model. It perturbs the input and observes how the predictions change, identifying the most influential parts of the input for the prediction. This is a saliency-based XAI method, visualizing the important regions in an image that the DNN relies on.

**SHAP (SHapley Additive exPlanations)**: SHAP is a unified approach to interpreting model predictions based on Shapley values from cooperative game theory. It assigns each feature an importance value for a particular prediction, offering a sound measure of feature importance. This is a saliency-based XAI method, visualizing the important regions in an image that the DNN relies on.

**GradCAM (Gradient-weighted Class Activation Mapping)**: GradCAM visualizes the regions in an image that contribute to the classification. It uses the gradients of the target concept (e.g., a specific class) flowing into the final convolutional layer to produce a coarse localization map highlighting important regions. This is a saliency-based XAI method.

**AblationCAM**: AblationCAM improves GradCAM by iteratively removing parts of the input and observing the output effect to identify important regions. This is a saliency-based XAI method, visualizing the crucial regions in an image that the DNN relies on.

Table 7: **XAI methods included in our dataset.** *Name* denotes the identifier of the utilized XAI method. *Functioning* specifies the mechanism of the explanation computation, including methods that rely on gradient weighting (Gradient), probing reactions to localized perturbations (Perturbation), abstracting activations through factorization (Factorization), leveraging directly interpretable latent spaces (Interpretable latent space), or searching for counterfactuals (Counterfactual). *Attribution* indicates the data type on which the attribution weights are applied: either on input images (Image) or on a computed representation of the image as concepts (Concepts). *Stage* indicates whether the explanation is produced by a ante-hoc or a post-hoc process.

| Name | Functioning | Attribution on | Stage | Applied on |
|---|---|---|---|---|
| BCos (Böhle et al., 2024) | Interpretable latent space | Image | Ante-hoc | ResNet50-BCos |
| GradCAM (Selvaraju et al., 2017) | Gradient | Image | Post-hoc | ViT, ResNet50, CLIP (zero-shot) |
| HiResCAM (Draelos & Carin, 2020) | Gradient | Image | Post-hoc | ViT, ResNet50, CLIP (zero-shot) |
| GradCAMElementWise (Pillai & Pirsiavash, 2021) | Gradient | Image | Post-hoc | ViT, ResNet50, CLIP (zero-shot) |
| GradCAM++ (Chattopadhay et al., 2018) | Gradient | Image | Post-hoc | ViT, ResNet50, CLIP (zero-shot) |
| XGradCAM (Fu et al., 2020) | Gradient | Image | Post-hoc | ViT, ResNet50, CLIP (zero-shot) |
| AblationCAM (Ramaswamy et al., 2020) | Perturbation | Image | Post-hoc | ViT, ResNet50, CLIP (zero-shot) |
| ScoreCAM (Wang et al., 2020) | Perturbation | Image | Post-hoc | ViT, ResNet50 |
| EigenCAM (Muhammad & Yeasin, 2020) | Factorization | Image | Post-hoc | ViT, ResNet50, CLIP (zero-shot) |
| EigenGradCAM (Muhammad & Yeasin, 2020) | Gradient+Factorization | Image | Post-hoc | ViT, ResNet50, CLIP (zero-shot) |
| LayerCAM (Jiang et al., 2021) | Gradient | Image | Post-hoc | ViT, ResNet50, CLIP (zero-shot) |
| FullGrad (Srinivas & Fleuret, 2019) | Gradient | Image | Post-hoc | ViT, ResNet50 |
| Deep Feature Factorizations (Collins et al., 2018) | Factorization | Image | Post-hoc | ViT, ResNet50, CLIP (zero-shot) |
| SHAP (Lundberg & Lee, 2017) | Perturbation | Image | Post-hoc | ViT, ResNet50, CLIP (zero-shot) |
| LIME (Ribeiro et al., 2016) | Perturbation | Image | Post-hoc | ViT, ResNet50, CLIP (zero-shot) |
| X-NeSyL (Díaz-Rodríguez et al., 2022) | Interpretable latent space | Concepts | Ante-hoc | X-NeSyL |
| CLIP-linear-sample (Yan et al., 2023) | Interpretable latent space | Concepts | Ante-hoc | CLIP-linear |
| CLIP-QDA-sample (Kazmierczak et al., 2024) | Counterfactual | Concepts | Ante-hoc | CLIP-QDA |
| LIME-CBM (Kazmierczak et al., 2024) | Perturbation | Concepts | Post-hoc | CLIP-QDA, ConceptBottleneck |
| SHAP-CBM (Kazmierczak et al., 2024) | Perturbation | Concepts | Post-hoc | CLIP-QDA, ConceptBottleneck |
| RISE (Petsiuk et al., 2018) | Perturbation | Concepts | Post-hoc | ConceptBottleneck |

**EigenCAM**: EigenCAM applies PCA to the activations of the last convolutional layer to produce a saliency map. It highlights the directions in which activations show the most variance, identifying critical features. This is a saliency-based XAI method.

**FullGrad**: FullGrad computes gradients of the output with respect to both the input and intermediate layer outputs, aggregating these gradients to generate a comprehensive saliency map. It is a saliency-based XAI method that visualizes key regions in an image.

**GradCAMPlusPlus**: GradCAMPlusPlus improves GradCAM with a refined weighting scheme for the gradients, allowing better handling of multiple occurrences of the target concept. This is a saliency-based XAI method.

**GradCAMElementWise**: GradCAMElementWise extends GradCAM by considering element-wise multiplications of gradients and activations, producing more precise visual explanations. This is a saliency-based XAI method.

**HiResCAM**: HiResCAM improves on class activation mapping by using higher-resolution feature maps for more detailed visual explanations. This is a saliency-based XAI method.

**ScoreCAM**: ScoreCAM improves CAM methods by using output scores to weight the activation maps' importance, providing a more faithful saliency map without relying on gradients. This is a saliency-based XAI method.

**XGradCAM**: XGradCAM integrates cross-layer information to combine saliency maps from different layers, producing a more comprehensive explanation. This is a saliency-based XAI method.

**DeepFeatureFactorization**: This method decomposes feature representations learned by a deep model into interpretable factors. It provides insights into how features contribute to the model's decisions, being a saliency-based XAI method.

**CLIP-QDA-sample**: This model uses the CLIP framework and applies Quadratic Discriminant Analysis (QDA) for classification. It links visual and concept-based representations to provide interpretable explanations. This is a concept bottleneck model (CBM).

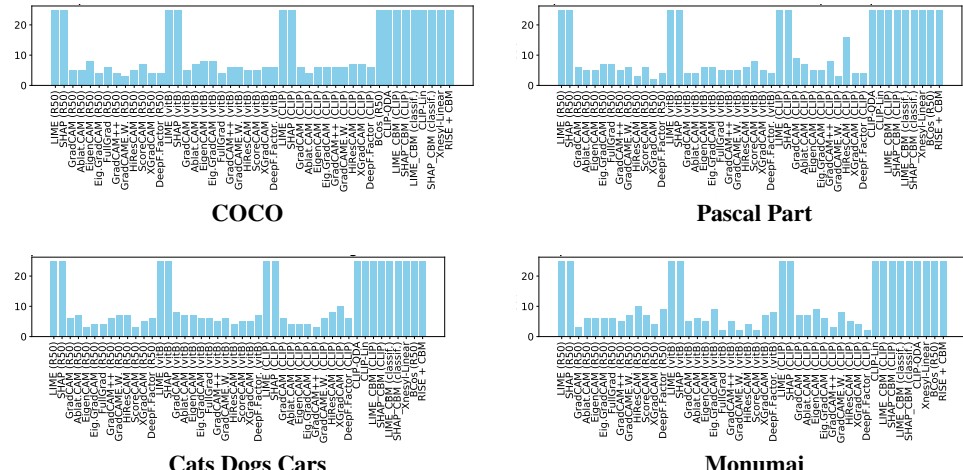

Figure 7: **Distribution across the different XAI techniques across the different datasets.**

**CLIP-Linear-sample**: Similar to CLIP-QDA, this model leverages the CLIP framework but applies logistic regression for classification, providing interpretable explanations based on concept representations. This is a CBM.

**X-NeSyL**: X-NeSyL identifies concepts using object detection and applies a small DNN to these concepts, using the weights assigned to each concept for explanation.

**LIME CBM**: This model generates a list of concepts and applies logistic regression. It uses LIME to highlight the most important concepts for classification.

**SHAP CBM**: This model generates a list of concepts and applies logistic regression, using SHAP to emphasize the most crucial concepts in classification.

**Labo**: Labo extracts human-interpretable concepts and maps them to the model's internal representations for more comprehensible decision-making explanations.

**RISE (Randomized Input Sampling for Explanation)**: RISE generates heatmaps by perturbing input regions and measuring their impact on model outputs. This technique identifies the most influential regions in the model's decision-making process.

**BCos**: BCos introduces specific layers to encourage alignment between weights and activation maps, which can then be used for explainability.

Finally, Figure 7 shows the distribution of XAI techniques applied across the datasets. To enhance the generalizability of our results, we increased the diversity of XAI techniques used. This was achieved by not applying every technique to every image uniformly, allowing for a more diverse set of explanations to be generated. This variability ensures that our analysis captures a broad spectrum of interpretability techniques, providing deeper insights into the performance of XAI techniques across different datasets and models.

### A.3 DATASET ANNOTATION PROCESS

The annotation process took place via an online web application, created and deployed by a contracting company. 15 participants were recruited to take part in the annotation process. These participants ranged in age from 19 to 37 (mean age 25.9, standard deviation 5.5). Figure 8 shows the age distribution. Among the participants, 5 identified themselves as male, 10 as female, 0 as non-binary, 0 did not wish to say. All participants were based in India.

Each participant's task was to annotate 147 explanations. For each image, participants had to explain what led them to classify the displayed image as the model's prediction. Participants responded openly using a text form. Similarly, they were asked to describe the elements of the image that helped them make the decision to classify the image as the model did. These two questions (Q0.1

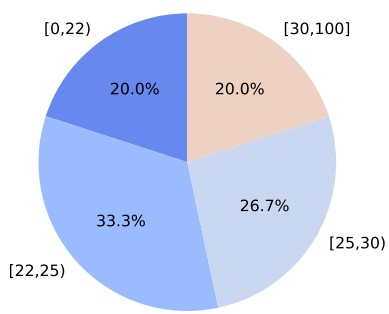

Figure 8: **Age distribution of the annotators.**

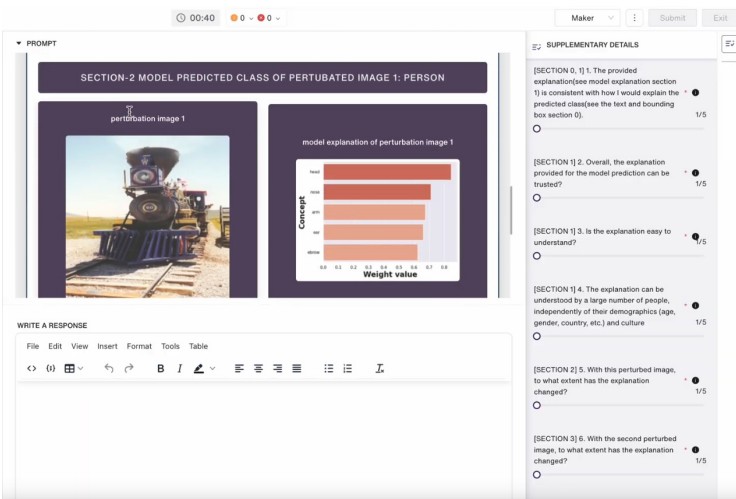

Figure 9: **Screenshot of the annotation interface.** Questions are on the right-part of the interface. Middle panel shows Section 2: slightly disturbed original image with the explanation.

and Q0.2) are used to establish a baseline for interpreting what makes an image recognizable as a specific object or class, and what are the salient features of the images that would explain this choice.

Once they had answered these two questions, they moved on to the next screen, where they had to answer a series of 6 questions (see Figure 9 showing a screenshot of the interface with the original image slightly perturbed). The questions concerned different sections of the screen. The first four questions (Q1 to Q4) concerned Section 1, which showed the original image on the left and the explanation on the right. These first four questions enabled participants to assess the levels of reliability, complexity and objectivity of the explanation. The fifth question (Q5) concerned Section 2, showing the slightly disturbed original image and the corresponding explanation. The sixth question (Q6) concerned Section 3, showing a more disturbed image and the corresponding explanation. These last two questions were intended to assess the robustness of the XAI technique. For each question, participants had to answer with a 5-point Likert scale.

All questions were formulated in collaboration with psychologists to enhance the quality of human feedback. Our aim was to avoid influencing responses and to eliminate any ambiguities that could lead to inaccurate answers. Human decision-making, especially when it involves assessing the quality of explanations, is complex. To address this, we provided training on deep learning and various XAI techniques, ensuring the content was clearly understandable by the annotators. After the initial training, the annotators answered the questions, and we held weekly meetings to clarify any confusion they encountered

## A.4 Perturbations

To address questions Q5 and Q6, which involve image perturbations, we outline the perturbation process below. To capture a diverse range of model responses, we applied 12 distinct types of perturbations, each with a tunable magnitude parameter to adjust perturbation intensity. The transformations include both standard torchvision operations `https://pytorch.org/vision/0.9/transforms.html` and custom-designed modifications:

- **Color Jitter (Brightness)**: Adjusts the brightness of the image. The magnitude lower the brightness.
- **Color Jitter (Contrast)**: Modifies image contrast. The magnitude lower the contrast.
- **Random Resized Crop**: Performs a random crop and resize. The magnitude augments the scale of the crop.
- **Gaussian Blur**: Blurs the image using a Gaussian filter. The magnitude augments the values of the standard deviation.
- **Random Perspective**: Applies a perspective transformation. The magnitude augments the distortion scale.
- **Brightness Transform**: Independently changes brightness levels. The magnitude lower the brightness.
- **Color Transform**: Adjusts color balance. The magnitude augments the saturation.
- **Contrast Transform**: Further modifies contrast. The magnitude augments the contrast.
- **Sharpness Transform**: Changes image sharpness. The magnitude augments the sharpness factor.
- **Posterize Transform**: Reduces color depth. The magnitude augments the number of bits to keep for each channel.
- **Solarize Transform**: Inverts colors above a certain threshold. The magnitude augments the threshold.
- **Random Masking**: Masks out random sections of the image by applying patches. The magnitude augments the number of patches.

## B  PASTA-dataset: additional experiments

### B.1  Comparison with existing benchmarks

We compiled a set of related works that perform human assessment in the context of XAI. Specifically, we noted key details such as the dataset size, the number of participants involved, the diversity of questions posed, and the overall scope of the study, where this information was available. A summary of these details is presented in Table 8.

The PASTA-dataset distinguishes itself in several key aspects. First, it evaluates a significantly higher number of XAI methods compared to existing datasets. This design emphasizes the diversity of techniques over the number of samples tested, offering a complementary approach to datasets that prioritize varied input data but evaluate fewer methods. Second, PASTA involves the lowest number of participants among the datasets listed, allowing for reduced variability due to potential outliers and better control over annotator behavior. However, this could introduce inherent biases tied to the limited participant pool. Finally, the PASTA-dataset provides a substantially larger number of samples and uniquely combines image-based explanations (e.g., saliency maps) with concept-based explanations (e.g., CBMs), making it the first dataset to address both modalities simultaneously.

### B.2  Comparison with existing metrics

Evaluating the quality of an explanation typically involves estimating different and potentially orthogonal aspects of it. In addition to the perceptual quality addressed in this work, others can be

Table 8: **Overview of datasets and human evaluation frameworks for XAI methods. Name** refers to the reference of the dataset used. *Annotations* refers to the type of labels used: Likert refers to Likert scale, Saliency refers to pseudo saliency maps, 2AFC refers to two alternative forced choices, Clictionary refers to the clicktionary game defined in (Dawoud et al., 2023), MCQ refers to multiple-choice question, and Binary refers to binary choises. $N_{Samples}$ refers to the total number of samples constituting the dataset. $N_{Part}$ refers to the number of participants involved during the labeling. *Modality* refers to the different modalities the dataset deals with: I refers to image, C to concepts, and T to text. $N_Q$ refers to the number of different questions asked to the annotators. $N_{XAI}$ refers to the number of XAI methods tested during the experiments, *No* indicates that the dataset only asked to label data with what they consider as ground truth explanation, without further comparison with XAI methods. $N_{Data}$ refers to the number of different samples (for example images) shown to annotators.

| **Name** | *Annotations* | $N_{Samples}$ | $N_{Part}$ | *Modality* | $N_Q$ | $N_{XAI}$ | $N_{Data}$ |
|---|---|---|---|---|---|---|---|
| PASTA-dataset | Likert | 66,000 | 15 | I + C | 6 | 21 | 100 |
| Yang et al. (2022) | Saliency, 2AFC | 356 | 46 | I | 2 | 1 | 89 |
| Colin et al. (2022) | Classification | 1,960 | 241 | I | 1 | 6 | NA |
| Dawoud et al. (2023) | Clicktionary | 3,836 | 76 | I | 1 | 3 | 102 |
| Mohseni et al. (2021) | Saliency | 1,500 | 200 | I + T | 1 | No | 1,500 |
| Herm et al. (2021) | Likert | NA | 165 | C | 1 | 6 | NA |
| Morrison et al. (2023) | Clicktionary/QCM | 450 | 50 | I | 1 | 3 | 39 |
| Spreitzer et al. (2022) | Likert/Binary | 4,050 | 135 | C | 9 | 2 | NA |
| Xuan et al. (2023) | Likert/Binary | 3,600 | 200 | C | 4 | 2 | 1,326 |

numerically simulated by having access to model weights. In this additional analysis, we consider some of those aspects and measure how much they correlate with human scores. The results cover only image-level attribution methods (see Table 7), as CBMs do not support such kinds of input-level manipulations.

***Faithfulness*:** *How much does the explanation describe the true behavior of the model?* A number of different ways to compute *faithfulness* exist, but they all broadly fit the same framework of measuring how much model predictions change in response to input perturbations (Bhatt et al., 2021; Alvarez-Melis & Jaakkola, 2018a; Yeh et al., 2019; Rieger & Hansen, 2020; Arya et al., 2019; Nguyen & Martínez, 2020; Bach et al., 2015; Samek et al., 2016; Montavon et al., 2018; Ancona et al., 2017; Dasgupta et al., 2022). Intuitively, an explanation is faithful if perturbing regions deemed irrelevant by the explanation bring little to no change in model output, whereas perturbing regions deemed relevant bring a considerable change. In this analysis, we resort to the evaluation protocol outlined in Azzolin et al. (2024), which generalized a number of common *faithfulness* metrics into a common mold[1]. Specifically, *faithfulness* is estimated as the harmonic mean of **sufficiency** (Suf) and **necessity** (Nec), which account for the degree of prediction changes after perturbing irrelevant or relevant portions of the input, respectively. Formally, given an input image $\boldsymbol{x}$ with associated explanation $\boldsymbol{e}$, and a model to be explained $p_\theta(Y \mid \boldsymbol{x})$, sufficiency and necessity are defined as:

$$\mathsf{Suf}_{d,p_R}(\boldsymbol{x}, \boldsymbol{e}) = \mathbb{E}_{\boldsymbol{x}' \sim p_R}[d(p_\theta(\cdot \mid \boldsymbol{x}) \parallel p_\theta(\cdot \mid \boldsymbol{x}'))] \tag{4}$$

$$\mathsf{Nec}_{d,p_C}(\boldsymbol{x}, \boldsymbol{e}) = \mathbb{E}_{\boldsymbol{x}' \sim p_C}[d(p_\theta(\cdot \mid \boldsymbol{x}) \parallel p_\theta(\cdot \mid \boldsymbol{x}'))],$$

where $d$ is a divergence between distributions of choice, and $p_C$ and $p_R$ are interventional distributions specifying the set of allowed perturbations to the explanation and its complement, respectively. Eq. 4 are then normalised to $[0, 1]$, the higher the better, via a non-linear transformation, i.e., taking $\exp(-\mathsf{Suf}_{d,p_R}(\boldsymbol{x}, \boldsymbol{e}))$ and $1 - \exp(-\mathsf{Nec}_{d,p_C}(\boldsymbol{x}, \boldsymbol{e}))$. Operationally, for a given instance $(\boldsymbol{x}, \boldsymbol{e})$ sampling from $p_C(\boldsymbol{x}, \boldsymbol{e})$ equals to generating a new image where the complement of the explanation is left intact, and where perturbations are applied to the explanation. The set of allowed perturbations $p_C$ and $p_R$ can be arbitrarily defined, and different techniques are oftentimes reported to give different interpretations (Hase et al., 2024; Rong et al., 2022). To avoid this confounding effect, we report the results for three different baseline perturbations, namely uniform and Gaussian noise, and black patches, along with a more advanced information-theoretic strategy named ROAD (Rong

---

[1]They focus on *faithfulness* for graph explanations, but the evaluation protocol is aligned with that of images.

et al., 2022). Since explanations are oftentimes in the form of soft relevance scores over the entire input, a threshold is needed to tell apart relevant from irrelevant image regions. To avoid relying upon this hard-to-define hyperparameter, we aggregate the scores across multiple thresholds keeping only the best value. Therefore, for each explanation threshold value, pixels are sorted based on their relevance[2] and progressively perturbed until reaching the fixed threshold value, while leaving the others unchanged. For each of those samples, we evaluate the normalised Eq. 4 where $d$ is the absolute difference in class-predicted confidence between clean and perturbed images, i.e., $|p_\theta(\hat{y} \mid \boldsymbol{x}) - p_\theta(\hat{y} \mid \boldsymbol{x}')|$, and average across the number of perturbed pixel for each threshold value. This procedure is detailed in Algorithm 1.

---

**Algorithm 1** Pseudo code for computing sufficiency/necessity

---

**Require:** Image $\boldsymbol{x}$, explanation $\boldsymbol{e}$, and set of explanation-size thresholds $\mathcal{T}$.

1: values $= []$
2: **for** each threshold $t \in \mathcal{T}$ **do**
3:     **if** computing sufficiency **then**
4:         Sort pixels of $\boldsymbol{x}$ in **ascending** order of relevance scores from $\boldsymbol{e}$.
5:     **else**
6:         Sort pixels of $\boldsymbol{x}$ in **descending** order of relevance scores from $\boldsymbol{e}$.
7:     **end if**
8:     arr $\leftarrow []$
9:     **for** $i$ in range(start=1, end=t, step=2) **do**
10:         $x' \leftarrow$ Apply the specified perturbations to the first $i\%$ sorted pixels.
11:         Append $d = |p_\theta(\hat{y} \mid \mathbf{x}) - p_\theta(\hat{y} \mid \mathbf{x}')|$ to arr
12:     **end for**
13:     **if** computing sufficiency **then**
14:         Append $\exp(-\text{mean}(\text{arr}))$ to values
15:     **else**
16:         Append $1 - \exp(-\text{mean}(\text{arr}))$ to values
17:     **end if**
18: **end for**
19: **Output:** max(arr)

---

***Robustness:*** *Robustness* roughly refers to how stable the explanation is to small input perturbations. Different ways to estimate it exist (Alvarez-Melis & Jaakkola, 2018b; Montavon et al., 2018; Yeh et al., 2019; Dasgupta et al., 2022; Agarwal et al., 2022a). In our analysis, we focused on MaxSensitivity (Yeh et al., 2019), which applies random input perturbations to the entire image and measures the pixel-wise difference between the original explanation, and the one obtained on the perturbed sample. Formally:

$$\text{MaxSensitivity} = \max\|\boldsymbol{e} - \boldsymbol{e}'\| \tag{5}$$

where $\boldsymbol{e}$ and $\boldsymbol{e}'$ are the explanations for the original and the perturbed image, respectively. Again, different perturbation techniques can be applied, and we resort to the two simple baselines, namely Uniform and Gaussian noise. No normalization is applied, therefore the values are the higher the worse. More advanced techniques like ROAD (Rong et al., 2022) cannot be applied in this context, since the perturbation is applied uniformly over the entire image. In Table 9, we report the correlation between MaxSensitivity and human scores, outlining a non-significant correlation with the metric and some questions. Surprisingly, the most correlated questions are **Q1-4**, which are not requesting humans to assess the stability of the explanation, something instead partially addressed by **Q5** and **Q6**. However, the correlation is very weak anyway, questioning any further claims.

***Complexity:*** As humans have an implicit tendency to favor simple alternatives when facing a comparison between different hypotheses, providing simple and compact explanations is vital for human-machine synergy (Cowan, 2001). Alternative methods for estimating the *complexity* of an explanation are available, from simple above-threshold counting to more advanced information-theoretic techniques (Chalasani et al., 2020; Bhatt et al., 2021; Nguyen & Martínez, 2020). To test whether those metrics are correlated to human scores, we report in Table 10 the correlation between

---

[2]For sufficiency, pixels are sorted in ascending order. For necessity, in descending order.

Table 9: **Pearson Correlation Coefficient (PCC) and Spearman rank Correlation Coefficient (SCC) between MaxSensitivity computed with different perturbation strategies and human scores.** In parentheses, the respective p-values.

|  | Uniform noise | | Gaussian noise | |
|---|---|---|---|---|
|  | PCC | SCC | PCC | SCC |
| **Q1** | -0.30 (1e-5) | -0.37 (1e-5) | -0.30 (1e-5) | -0.35 (0.04) |
| **Q2** | -0.30 (1e-5) | -0.36 (1e-5) | -0.28 (1e-5) | -0.33 (0.17) |
| **Q3** | -0.30 (1e-5) | -0.36 (1e-5) | -0.29 (1e-5) | -0.34 (0.16) |
| **Q4** | -0.29 (1e-5) | -0.36 (1e-5) | -0.28 (0.19) | -0.33 (0.33) |
| **Q5** | 0.01 (0.72) | 0.01 (0.80) | -0.04 (0.89) | -0.09 (0.01) |
| **Q6** | 0.10 (1e-3) | 0.10 (1e-3) | 0.09 (0.01) | 0.10 (2e-3) |

Table 10: **Pearson Correlation Coefficient (PCC) and Spearman rank Correlation Coefficient (SCC) between Sparseness and human scores.** In parentheses the respective p-values.

|  | *Complexity* | |
|---|---|---|
|  | PCC | SCC |
| **Q1** | -0.17 (1e-10) | -0.15 (1e-8) |
| **Q2** | -0.17 (1e-10) | -0.15 (1e-8) |
| **Q3** | -0.15 (1e-9) | -0.13 (1e-7) |
| **Q4** | -0.15 (1e-7) | -0.13 (1e-6) |
| **Q5** | -0.21 (1e-15) | -0.22 (1e-17) |
| **Q6** | 0.02 (0.45) | 0.05 (0.09) |

human votes and Sparseness (Chalasani et al., 2020), which estimates explanation *complexity* as the Gini Index (Hurley & Rickard, 2009) of the absolute values of the image attribution. The result is a metric value in the range $[0, 1]$, where higher values indicate more sparseness. The computation of the Gini index is detailed in Algorithm 2.

---

**Algorithm 2** Pseudo code for Gini coefficient calculation from Hedström et al. (2023)

---

**Require:** Explanation $e$
1: $array \leftarrow flatten(e)$
2: $array \leftarrow |array|$ {Take absolute value}
3: $array \leftarrow sort(array, ascending = True)$
4: $index \leftarrow arange(1, array.shape[0] + 1)$
5: $n \leftarrow array.shape[0]$
6: **return** $\frac{\sum(2 \cdot index - n - 1) \cdot array}{n \cdot \sum array}$

---

We used the Quantus library (Hedström et al., 2023) for implementing the previous metrics, and we present the raw metric values in Table 11, aggregated by explainer and model. Overall, none of the above metrics exhibit a significant correlation with user scores.

### B.3 DATASET ANALYSIS

To thoroughly analyze the dataset and evaluate potential biases, we conducted several tests. First, we experimented with various aggregation techniques, ultimately selecting the majority voting method as the most effective. To further explore annotator preferences, we identified the top-12 XAI techniques selected by each annotator and visualized the results in the histogram shown in Figure 10. From this figure, we observe that classical methods such as LIME and SHAP stand out as the most frequently preferred. This suggests a strong preference for well-established saliency-based methods. Additionally, there is a notable inclination towards methods that probe the model's reaction to input perturbations. A distinct aspect is the inclusion of B-cos, which generates explanations through the incorporation of a dedicated layer, offering a unique mechanism compared to other perturbation-based techniques. Of the 11 most popular techniques among annotators, only four are

Table 11: **Raw metric values averaged for each explainer and model.** Each value is the average result on 5 runs with the standard deviation.

| Saliency Method | Model | Faithfulness (ROAD) | Robustness (Gaussian noise) | Sparseness |
|---|---|---|---|---|
| GradCAM | resnet50 | $0.08 \pm 0.16$ | $0.79 \pm 0.45$ | $0.68 \pm 0.11$ |
| GradCAM | vitB | $0.02 \pm 0.04$ | $1.11 \pm 0.96$ | $0.41 \pm 0.19$ |
| AblationCAM | vitB | $0.02 \pm 0.02$ | $1.83 \pm 1.68$ | $0.77 \pm 0.23$ |
| AblationCAM | CLIP-zero-shot | $0.01 \pm 0.01$ | $1.46 \pm 0.44$ | $0.83 \pm 0.15$ |
| EigenCAM | resnet50 | $0.11 \pm 0.19$ | $0.94 \pm 0.46$ | $0.80 \pm 0.05$ |
| EigenCAM | vitB | $0.02 \pm 0.03$ | $0.98 \pm 0.35$ | $0.59 \pm 0.06$ |
| EigenCAM | CLIP-zero-shot | $0.02 \pm 0.02$ | $0.76 \pm 0.37$ | $0.54 \pm 0.08$ |
| EigenGradCAM | resnet50 | $0.11 \pm 0.18$ | $0.94 \pm 0.70$ | $0.78 \pm 0.10$ |
| EigenGradCAM | vitB | $0.02 \pm 0.03$ | $2.10 \pm 1.38$ | $0.88 \pm 0.11$ |
| EigenGradCAM | CLIP-zero-shot | $0.03 \pm 0.02$ | $1.55 \pm 1.27$ | $0.73 \pm 0.17$ |
| FullGrad | resnet50 | $0.05 \pm 0.14$ | $0.43 \pm 0.06$ | $0.43 \pm 0.07$ |
| FullGrad | vitB | $0.02 \pm 0.02$ | $1.30 \pm 0.46$ | $0.39 \pm 0.06$ |
| GradCAM | CLIP-zero-shot | $0.03 \pm 0.03$ | $1.29 \pm 0.65$ | $0.60 \pm 0.16$ |
| GradCAMPlusPlus | resnet50 | $0.08 \pm 0.13$ | $0.62 \pm 0.20$ | $0.63 \pm 0.09$ |
| GradCAMPlusPlus | vitB | $0.02 \pm 0.03$ | $2.45 \pm 1.90$ | $0.69 \pm 0.24$ |
| GradCAMPlusPlus | CLIP-zero-shot | $0.01 \pm 0.01$ | $1.53 \pm 0.97$ | $0.65 \pm 0.23$ |
| GradCAMElementWise | resnet50 | $0.07 \pm 0.16$ | $0.70 \pm 0.34$ | $0.58 \pm 0.08$ |
| GradCAMElementWise | vitB | $0.02 \pm 0.03$ | $1.17 \pm 0.37$ | $0.61 \pm 0.11$ |
| GradCAMElementWise | CLIP-zero-shot | $0.03 \pm 0.03$ | $0.73 \pm 0.16$ | $0.38 \pm 0.07$ |
| HiResCAM | resnet50 | $0.12 \pm 0.19$ | $0.86 \pm 0.32$ | $0.65 \pm 0.11$ |
| HiResCAM | vitB | $0.02 \pm 0.02$ | $1.71 \pm 0.83$ | $0.70 \pm 0.22$ |
| HiResCAM | CLIP-zero-shot | $0.02 \pm 0.03$ | $1.45 \pm 0.41$ | $0.68 \pm 0.13$ |
| LIME | resnet50 | $0.13 \pm 0.20$ | $0.50 \pm 0.07$ | $0.12 \pm 0.04$ |
| ScoreCAM | resnet50 | $0.07 \pm 0.17$ | $0.85 \pm 0.43$ | $0.56 \pm 0.09$ |
| ScoreCAM | vitB | $0.03 \pm 0.04$ | $1.24 \pm 0.80$ | $0.46 \pm 0.18$ |
| XGradCAM | resnet50 | $0.14 \pm 0.19$ | $1.00 \pm 0.41$ | $0.71 \pm 0.10$ |
| XGradCAM | vitB | $0.02 \pm 0.02$ | $1.32 \pm 0.16$ | $0.61 \pm 0.06$ |
| XGradCAM | CLIP-zero-shot | $0.03 \pm 0.03$ | $1.31 \pm 0.12$ | $0.58 \pm 0.04$ |
| DeepFeatureFactorization | resnet50 | $0.07 \pm 0.17$ | $1.28 \pm 0.54$ | $0.34 \pm 0.11$ |
| DeepFeatureFactorization | vitB | $0.02 \pm 0.03$ | $0.62 \pm 0.21$ | $0.28 \pm 0.05$ |
| DeepFeatureFactorization | CLIP-zero-shot | $0.03 \pm 0.05$ | $0.73 \pm 0.36$ | $0.36 \pm 0.09$ |
| LIME | vitB | $0.02 \pm 0.03$ | $0.50 \pm 0.05$ | $0.15 \pm 0.03$ |
| BCos | resnet50-bcos | $0.17 \pm 0.25$ | $0.85 \pm 0.32$ | $0.50 \pm 0.09$ |
| LIME | CLIP-zero-shot | $0.03 \pm 0.04$ | $0.61 \pm 0.10$ | $0.16 \pm 0.05$ |
| SHAP | resnet50 | $0.12 \pm 0.20$ | $1.38 \pm 0.55$ | $0.50 \pm 0.11$ |
| SHAP | vitB | $0.02 \pm 0.03$ | $1.05 \pm 0.48$ | $0.40 \pm 0.10$ |
| SHAP | CLIP-zero-shot | $0.02 \pm 0.03$ | $1.31 \pm 0.52$ | $0.48 \pm 0.11$ |
| AblationCAM | resnet50 | $0.12 \pm 0.22$ | $0.95 \pm 0.48$ | $0.59 \pm 0.12$ |

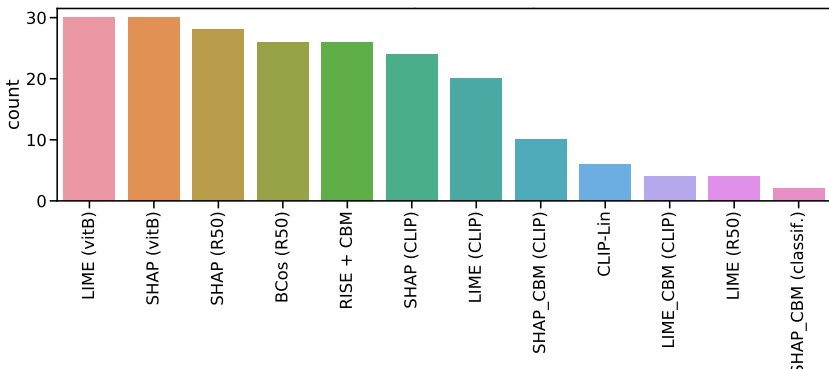

Figure 10: **Histogram showing the top-12 XAI techniques preferred by each annotator.**

based on Concept Bottleneck Models (CBMs), indicating a general preference for saliency maps over concept-based explanations. Unlike Table 12, this analysis focuses on the top-12 techniques per annotator, removing the influence of votes among the top-12 techniques to reduce noise and better capture annotator preferences.

Next, we aggregated the scores using majority voting and calculated QWC scores to measure the agreement between individual annotators and the aggregated score. We further analyzed the QWC scores by gender and age groups to assess any systematic differences in interpretation. As shown in Figure 12, the kappa scores indicate that there is generally consistent agreement across different age and sex groups, although older annotators show slightly less consistency. This highlights that while demographic factors may introduce some variation, they do not substantially impact the overall interpretability evaluation.

We also investigated potential biases in the annotations themselves by examining the differences in how annotators approached CBM-based and saliency-based explanations. For CBM explanations, we focused on the text written by annotators in response to question Q0.1, assessing whether annotators preferred explanations that closely resembled their own textual responses. To quantify this, we transformed CBM explanations into text by concatenating the top concepts used in the explanation and calculated the BLEU (Papineni et al., 2002) and ROUGE scores (Lin, 2004) between these explanations and the annotators' text responses. As shown in Figure 11, the ROUGE score reveals a slight correlation between the explanations and the annotators' expectations for questions Q1, Q2, Q3, and Q4. This suggests that annotators are inclined to favor explanations that align with their preconceived notions, potentially introducing a bias toward consistency with their initial answers.

Moreover, we observed that questions Q1, Q2, Q3, and Q4 exhibit high intercorrelation, as do questions Q5 and Q6. This clustering indicates that annotators tend to evaluate explanations similarly across these sets of questions, which may reflect underlying patterns in how different types of explanations are perceived.

For saliency-based explanations, we analyzed the bounding boxes provided by annotators in response to question Q0.2. We evaluated the correlation between various metrics and the annotators' answers, including:

1. the total sum of pixel intensities in the saliency map ("SUM_all"),

2. the sum of pixel intensities within the area of the image identified by the bounding box ("SUM_pos"),

3. the sum of pixel intensities outside the bounding box ("SUM_neg"),

4. the entropy of the saliency map ("Entropy").

Figure 11 shows that questions Q1, Q2, Q3, and Q4 are highly correlated with each other, as are questions Q5 and Q6. Additionally, all metrics except for "SUM_pos" show some correlation with questions Q1–Q4. This suggests that annotators may focus heavily on background features and salient objects when answering these questions, potentially overlooking finer details in the bounding box area.

Overall, these analyses highlight several potential biases in the dataset. Annotators exhibit a preference for certain types of explanations, particularly saliency maps, and tend to favor explanations that align with their expectations, as evidenced by the correlation between their text responses and the explanations. Additionally, while demographic factors such as age and gender do not significantly impact the overall evaluation, the slight decrease in consistency among older annotators warrants further investigation. The study involved 15 annotators, all from the same cultural background, which may introduce some shared perspectives or biases. To mitigate this, future studies could benefit from a more diverse group of annotators.

### B.4 ADDITIONAL RESULTS OF THE HUMAN EVALUATIONS

In Table 12, we present the average mode of votes for each XAI technique. the first observation is that it is difficult to observe clear differences among XAI methods. This is mainly due to the fact XAI methods are highly sensible to the backbone they are applied on, as noticed in Section 3.5. We observe also that the average score for saliency-based techniques across *fidelity* related questions is 2.47, while for CBMs, the average score is lower at 2.12. For saliency-based techniques across *complexity* related questions is 2.44, while for CBMs, the average score is also lower at 2.10. For saliency-based techniques across *objectivity* related questions is 2.46, while for CBMs, the average score is also lower at 2.11. For *robustness* related questions, the average scores are 3.59 for saliency-based techniques and 3.20 for CBMs.

Table 12: **XAI techniques with aggregated scores across different evaluation metrics.**

| XAI Technique | Fidelity (Q1) | Complexity (Q2-Q3) | Objectivity (Q4) | Robustness (Q5-Q6) |
|---|---|---|---|---|
| GradCAM (ResNet50) | 2.50 | 2.53 | 2.60 | 3.83 |
| GradCAM (ViT-B) | 2.77 | 2.86 | 3.00 | 3.57 |
| GradCAM (CLIP-zero-shot) | 1.86 | 1.78 | 1.76 | 3.90 |
| LIME (ResNet50) | 2.16 | 2.08 | 2.13 | 4.28 |
| LIME (ViT-B) | 3.06 | 2.98 | 2.99 | 3.89 |
| LIME (CLIP-zero-shot) | 2.48 | 2.49 | 2.51 | 4.25 |
| SHAP (ResNet50) | 2.63 | 2.66 | 2.65 | 3.74 |
| SHAP (ViT-B) | 2.87 | 2.85 | 2.89 | 3.64 |
| SHAP (CLIP-zero-shot) | 2.55 | 2.52 | 2.47 | 3.70 |
| AblationCAM (ResNet50) | 2.78 | 2.96 | 2.91 | 3.14 |
| AblationCAM (ViT-B) | 1.75 | 1.75 | 1.75 | 3.84 |
| AblationCAM (CLIP-zero-shot) | 1.27 | 1.34 | 1.36 | 3.64 |
| EigenCAM (ResNet50) | 2.23 | 2.23 | 2.41 | 3.21 |
| EigenCAM (ViT-B) | 3.31 | 3.25 | 3.15 | 3.39 |
| EigenCAM (CLIP-zero-shot) | 3.68 | 3.68 | 3.64 | 3.28 |
| EigenGradCAM (ResNet50) | 2.81 | 2.91 | 3.10 | 3.24 |
| EigenGradCAM (ViT-B) | 1.41 | 1.21 | 1.21 | 3.86 |
| EigenGradCAM (CLIP-zero-shot) | 2.29 | 2.10 | 2.05 | 3.67 |
| FullGrad (ResNet50) | 3.65 | 3.68 | 3.65 | 3.26 |
| FullGrad (ViT-B) | 1.65 | 1.50 | 1.65 | 4.00 |
| GradCAMPlusPlus (ResNet50) | 2.55 | 2.60 | 2.60 | 3.45 |
| GradCAMPlusPlus (ViT-B) | 2.19 | 2.00 | 1.90 | 3.93 |
| GradCAMPlusPlus (CLIP-zero-shot) | 1.80 | 1.88 | 1.85 | 4.13 |
| GradCAMElementWise (ResNet50) | 3.04 | 2.78 | 2.74 | 3.59 |
| GradCAMElementWise (ViT-B) | 1.47 | 1.53 | 1.53 | 3.90 |
| GradCAMElementWise (CLIP-zero-shot) | 2.39 | 2.33 | 2.35 | 3.24 |
| HiResCAM (ResNet50) | 2.96 | 2.77 | 2.83 | 3.52 |
| HiResCAM (ViT-B) | 1.45 | 1.30 | 1.30 | 4.10 |
| HiResCAM (CLIP-zero-shot) | 1.83 | 1.79 | 1.92 | 3.85 |
| ScoreCAM (ResNet50) | 2.68 | 2.55 | 2.45 | 3.36 |
| ScoreCAM (ViT-B) | 3.00 | 3.00 | 3.00 | 3.63 |
| XGradCAM (ResNet50) | 2.57 | 2.72 | 2.86 | 3.50 |
| XGradCAM (ViT-B) | 2.00 | 2.18 | 2.18 | 4.02 |
| XGradCAM (CLIP-zero-shot) | 2.50 | 2.31 | 2.21 | 4.34 |
| DeepFeatureFactorization (ResNet50) | 3.50 | 3.40 | 3.33 | 3.25 |
| DeepFeatureFactorization (ViT-B) | 2.46 | 2.66 | 2.69 | 3.45 |
| DeepFeatureFactorization (CLIP-zero-shot) | 2.94 | 2.92 | 3.22 | 3.61 |
| BCos (ResNet50-BCos) | 2.91 | 2.84 | 2.77 | 3.34 |
| CLIP-QDA-sample | 1.71 | 1.69 | 1.66 | 4.04 |
| CLIP-Linear-sample | 2.19 | 2.17 | 2.27 | 2.89 |
| LIME_CBM (CLIP-QDA) | 2.22 | 2.20 | 2.25 | 4.27 |
| SHAP_CBM (CLIP-QDA) | 2.44 | 2.33 | 2.29 | 3.81 |
| LIME_CBM (CBM-classifier-logistic) | 1.66 | 1.66 | 1.65 | 3.53 |
| SHAP_CBM (CBM-classifier-logistic) | 1.98 | 2.01 | 1.94 | 3.30 |
| Xnesyl-Linear | 1.72 | 1.77 | 1.77 | 3.82 |
| RISE (CBM-classifier-logistic) | 2.65 | 2.57 | 2.62 | 2.70 |

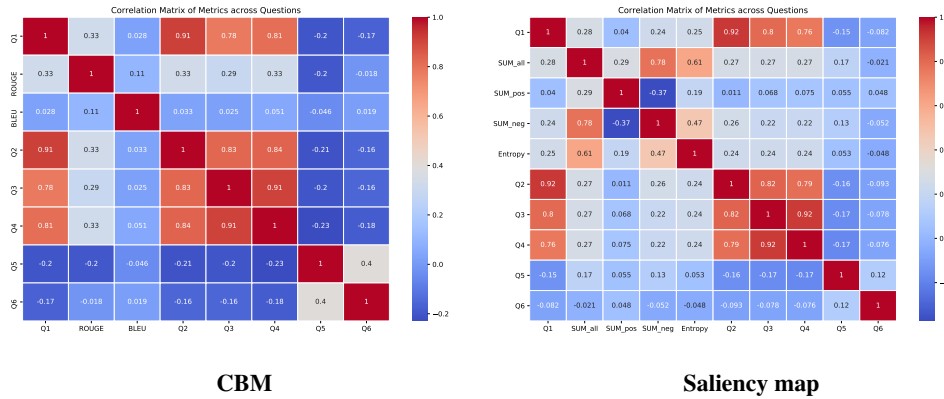

**CBM**         **Saliency map**

Figure 11: **Correlation between the various questions and key metrics for saliency and CBM explanations.** For CBM, the criteria used are the BLEU (Papineni et al., 2002) and ROUGE scores (Lin, 2004) scores between the explanation and the text from question Q0.1. For saliency maps, the metrics include the total pixel sum (SUM_all), the sum of pixels within the annotator-provided bounding box (SUM_pos) from question Q0.2, the sum of pixels outside the bounding box (SUM_neg), and the entropy of the saliency map (Entropy).

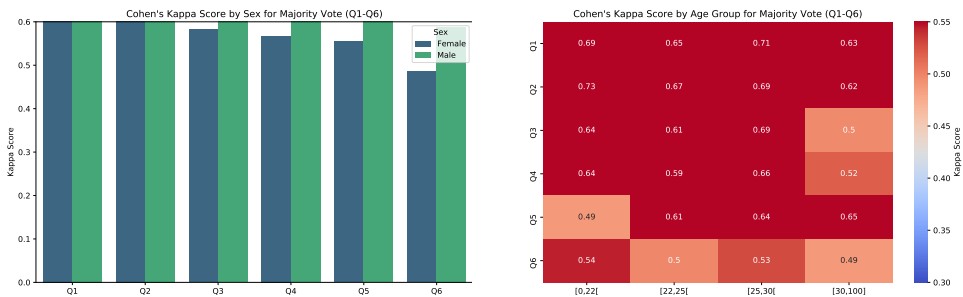

Figure 12: **Cohen's kappa statistics showing agreement between annotators, aggregated by sex and age groups.**

### B.5 EVALUATION QUESTIONS

In addition to annotating the samples that comprise the PASTA-dataset, each participant in the study was asked to respond to the following questions:

- Q7: Could you rank the different qualities of the explanation in order of importance? *Expectedness*, *Trustworthy*, *Complexity*, *Robustness*, *Objectivity*.
- Q8: Can you please order the different questions from Q1, Q2, Q3, Q4, Q5, and Q6 from the less important to the more important questions to assess the quality of the explanation?
- Q9: Can you please order the different questions from Q1, Q2, Q3, Q4, Q5 and Q6 from the most difficult to the easiest?

The results for Q7 are presented in Table 13. The data indicate that evaluators place a higher value on Trustworthiness and Complexity, while Objectivity is ranked significantly lower. This finding aligns with the work of Liao et al. (2022), which poses a similar question but focuses on a pool of experts.

Table 14 summarizes the results for Q8 and Q9. According to user feedback, Q1 is deemed the most important question. Notably, there is a strong correlation between the responses to Q8 and Q9: the questions perceived as easiest to answer are also regarded as the most important. Interestingly, there

is a low correlation between the importance assigned to each question and the axioms they represent, highlighting a distinction between the perception of an axiom and the execution of the associated task. Furthermore, both Q8 and Q9 reveal a separation in ranking among Q1 to Q4 and Q5 and Q6, which resonates with the dataset analysis discussed in Section B.3.

Table 13: **Average positions of axioms for question 7.** Lower rankings indicate that the axiom is considered more important by evaluators, while higher rankings suggest the axiom is considered less important. Results sorted by ascending order.

| Axiom | Average Position (Q7) $\downarrow$ |
|---|---|
| Trustworthy | **2.47** $\pm$ 1.48 |
| Complexity | 2.53 $\pm$ 1.58 |
| Robustness | 2.97 $\pm$ 1.06 |
| Expectedness | 3.20 $\pm$ 1.05 |
| Objectivity | 3.77 $\pm$ 1.36 |

Table 14: **Average positions of questions 1 to 6 for Q8 and Q9.** For Q8, higher rankings indicate that the question is considered more important by evaluators, while lower rankings suggest the question is considered less important. For Q9, lower rankings indicate that the question is considered more difficult to evaluate, while lower rankings suggest the question is considered less difficult.

| Question | Average Position (Q8) $\uparrow$ | Average Position (Q9) $\downarrow$ |
|---|---|---|
| Q1 | **4.10** $\pm$ 1.32 | 4.00 $\pm$ 1.21 |
| Q2 | 3.90 $\pm$ 1.29 | 4.23 $\pm$ 1.35 |
| Q3 | 4.03 $\pm$ 1.77 | 4.53 $\pm$ 1.70 |
| Q4 | 3.13 $\pm$ 1.41 | 3.67 $\pm$ 1.40 |
| Q5 | 2.63 $\pm$ 1.66 | **1.80** $\pm$ 0.57 |
| Q6 | 3.00 $\pm$ 1.95 | 2.54 $\pm$ 1.72 |

## C  PASTA-METRIC

### C.1  IMPLEMENTATION DETAILS

#### C.1.1  AGGREGATION OF THE VOTES.

In the dataset, we have access to 5 votes per question. Then, if we denote the set of votes as $\{m_i^j[a]\}_{a=1}^{N_a}$, where $N_a = 5$ is the number of annotations:

$$\text{Mode}(m_i^j) = \arg\max_a \text{Count}(m_i^j[a]), \tag{6}$$

where $\text{Count}(m_i^j[a])$ represents the frequency of each vote $a$ in the set.

#### C.1.2  EVALUATION METRICS

The quadratic weighted Cohen's Kappa (QWK) measures inter-rater agreement, adjusting for chance and penalizing disagreement based on its magnitude. The formula is:

$$\text{QWK} = \frac{\sum_{i,j} w_{ij} O_{ij} - \sum_{i,j} w_{ij} E_{ij}}{1 - \sum_{i,j} w_{ij} E_{ij}}, \tag{7}$$

where:

- $O_{ij}$ and $E_{ij}$ are the observed and expected frequencies, respectively.

- $w_{ij} = 1 - \frac{(i-j)^2}{(k-1)^2}$ is the quadratic weight for categories $i$ and $j$.

Table 15: **Comparison of the influence of ground truth generation methods.** Each value is the average result on 5 runs with the standard deviation.

| Label Type | MSE | QWK | SCC |
|---|---|---|---|
| Mode | $1.06 \pm 0.05$ | $0.48 \pm 0.05$ | $\mathbf{0.25 \pm 0.25}$ |
| Mean | $\mathbf{0.75 \pm 0.06}$ | $0.46 \pm 0.06$ | $0.24 \pm 0.25$ |
| Median | $0.95 \pm 0.03$ | $\mathbf{0.49 \pm 0.04}$ | $\mathbf{0.25 \pm 0.26}$ |

Table 16: **Impact of Adding Labels to the Encodings.** Each value is the average result on 5 runs with the standard deviation.

| Computation | MSE | QWK | SCC |
|---|---|---|---|
| *Embeddings* | $1.06 \pm 0.05$ | $0.48 \pm 0.05$ | $0.25 \pm 0.25$ |
| *Embeddings + Labels* | $1.09 \pm 0.00$ | $0.43 \pm 0.00$ | $0.22 \pm 0.22$ |

The Mean Squared Error (MSE) measures the average squared difference between predicted and actual values. It is given by:

$$\text{MSE} = \frac{1}{N_s} \sum_{k=1}^{N_s} (m_k - \hat{m}_k)^2 \,. \tag{8}$$

The Spearman Correlation Coefficient (SCC) measures the rank correlation between two variables. It is calculated using the ranks of the data points and is given by:

$$\text{SCC} = 1 - \frac{6 \sum_i d_i^2}{N_s(N_s^2 - 1)}, \tag{9}$$

where $d_i$ is the difference between the ranks of each pair of values.

## C.2 AGGREGATION OF THE VOTES

Given the subjective nature of the annotations and the presence of multiple responses to the same question (five answers per question), we explored different methods for determining the ground truth. In Table 15, we tested how PASTA-metric training is affected when using the mean, mode, or median as the ground truth. Since this parameter significantly impacts the dispersion of the samples, it is not surprising that the results vary, particularly when using the mean. However, in all our experiments, we opted to use the mode, as phenomena of high non-consensus were observed (see Appendix B.3).

## C.3 ADD OF LABEL INFORMATION IN THE PASTA-METRIC EMBEDDING.

In the current version of the PASTA-metric, only the activation outputs—either the heatmaps for saliency maps or the concept scores for CBMs—are utilized, without incorporating additional information that could potentially enhance the scoring process. To address this, we propose integrating information about the predicted class into the embeddings provided to the scoring network. Specifically, we encode each predicted label as a one-hot vector and concatenate it with the embedding. The results of this modified approach are presented in Table 16.

As observed in Table 16, incorporating label information into our framework appears to degrade performance. This phenomenon may be attributed to the relatively high number of labels used across all datasets (26), which is comparable to the number of distinct images. Consequently, the added label information may introduce redundancy or overfitting, ultimately impacting the overall scoring process.

## C.4 SCORING FUNCTIONS

In this section, we examine the influence of various scoring network architectures on performance. Specifically, we tested alternatives such as Ridge Regression, Lasso Regression, Support Vector

Table 17: **Impact of the Scoring Function.** Each value is the average result on 5 runs with the standard deviation.

| Scoring Function | MSE | QWK | SCC |
|---|---|---|---|
| PASTA | $1.06 \pm 0.05$ | $\mathbf{0.48 \pm 0.05}$ | $\mathbf{0.25 \pm 0.25}$ |
| SVM | $\mathbf{0.97 \pm 0.06}$ | $0.39 \pm 0.05$ | $0.22 \pm 0.22$ |
| Ridge | $0.98 \pm 0.05$ | $0.37 \pm 0.05$ | $0.22 \pm 0.22$ |
| Lasso | $1.71 \pm 0.18$ | $0.31 \pm 0.02$ | $0.16 \pm 0.16$ |
| MLP | $1.28 \pm 0.10$ | $0.38 \pm 0.04$ | $0.20 \pm 0.20$ |

Machines, and a Multi-Layer Perceptron with a single hidden layer of 100 units. The results of these experiments are presented in Table 17.

By analyzing the performance of the different scoring functions, we observe that PASTA, implemented with linear regression and leveraging the loss functions described in Section 4.2, achieves superior results in terms of the Quadratic Weighted Kappa score and Spearman Correlation Coefficient. These outcomes highlight its effectiveness in accurately ranking labels. However, both SVM and Ridge Regression exhibit lower Mean Square Error, suggesting better numerical precision in predicting label values. Our primary objective is to develop a robust metric for ranking XAI methods. As such, we place greater emphasis on metrics that assess ranking accuracy. Based on this criterion, the PASTA framework is favored over alternative scoring networks due to its superior performance in rank-oriented evaluations.

## C.5 LOSS FUNCTIONS

Here we define the three losses used to train our PASTA-model.

The Cosine Similarity Loss measures the cosine similarity between the predicted explanations $\hat{m}_k$ and the ground truth explanations $m_k$, ensuring alignment in their direction:

$$L_s = 1 - \frac{\sum_{k=1}^{N_s} \hat{m}_k m_k}{\sqrt{\sum_{k=1}^{N_s} \hat{m}_k^2}\sqrt{\sum_{k=1}^{N_s} m_k^2}} \tag{10}$$

The Mean Squared Error (MSE) loss measures the squared difference between predicted and true explanations, penalizing larger errors more heavily:

$$L_{mse} = \frac{1}{N_s} \sum_{k=1}^{N_s} (\hat{m}_k - m_k)^2 \tag{11}$$

This Ranking Loss ensures the correct ranking of explanations by penalizing cases where the predicted ranking contradicts the true ranking:

$$L_r = \frac{1}{\hat{N}_s} \sum_{k_1, k_2} \max(0, -(\hat{m}_{k_1} - \hat{m}_{k_2})(m_{k_1} - m_{k_2})) \tag{12}$$

Here, $\hat{N}_s$ represents the total number of pairs considered for the ranking loss.

## C.6 EXPLANATION EMBEDDINGS

**Saliency** Regarding saliency-based explanations, a key question arises about what should be considered as the image representing the explanation. Two variants were considered: using the heatmap visualization that is presented to users, as shown in Equation 1, or the input image as defined as:

$$\phi_{\text{image}}(e_i^j) = \text{CLIP}_{\text{image}}(x_i \times e_i^j) \tag{13}$$

Table 18: **Influence of the saliency computation process.** *Heatmap* refers to the process defined in Equation 1 and *Masked image* refers to the process defined in Equation 13. Each value is the average result on 5 runs with the standard deviation.

| Embedded Image | MSE | QWK | SCC |
|---|---|---|---|
| Heatmap | $\mathbf{1.06 \pm 0.05}$ | $\mathbf{0.48 \pm 0.05}$ | $\mathbf{0.25 \pm 0.25}$ |
| Blur | $1.08 \pm 0.14$ | $0.47 \pm 0.07$ | $0.24 \pm 0.27$ |

Table 19: **Influence of the number of words selected as an input text $N_{top}$.** Each value is the average result on 5 runs with the standard deviation.

| $N_{top}$ | MSE | QWK | SCC |
|---|---|---|---|
| 5 | $1.16 \pm 0.13$ | $0.46 \pm 0.06$ | $0.24 \pm 0.24$ |
| 10 | $1.17 \pm 0.07$ | $0.45 \pm 0.04$ | $0.23 \pm 0.23$ |
| 15 | $1.12 \pm 0.07$ | $0.46 \pm 0.04$ | $0.24 \pm 0.24$ |
| 20 | $\mathbf{1.06 \pm 0.05}$ | $\mathbf{0.48 \pm 0.05}$ | $\mathbf{0.25 \pm 0.25}$ |
| 25 | $1.11 \pm 0.06$ | $0.47 \pm 0.06$ | $0.25 \pm 0.25$ |
| 30 | $1.10 \pm 0.05$ | $0.47 \pm 0.05$ | $0.24 \pm 0.25$ |

The element-wise multiplication of the input image with the saliency map selectively blurs the image, with regions corresponding to lower activation values being blurred, while areas with higher activation values remain clear.

The results are presented in Table 18, where a slight improvement is observed in favor of using the image as a heatmap. This can be attributed to the fact that, despite being more computationally ambiguous, the heatmap display reveals the entire image. Additionally, this representation closely resembles the format of the samples provided to annotators.

**CBM** Concerning concept bottleneck explanation, which basically can be interpreted as a dictionary attributing a scalar for each concept. One crucial step is converting CBM activations to CLIP embeddings. We tested two ways do do so:

- By considering the raw text of concepts, ordered by importance (Equation 2)

- By using a sum of all the CLIP embeddings of text, weightened by its activations:

$$\phi_{text}(\boldsymbol{e}_i^j) = \frac{1}{||\boldsymbol{e}_i^j||} \sum_k^{N_k} \boldsymbol{e}_i^j[k] \, CLIP_{text}(concept_i[k]) \tag{14}$$

If we use the first solution, there are questions about the number of concepts to keep, that we note as the parameters $N_{top}$. Table 19 presents the influence of $N_{top}$ while 20 presents the influence of differents ways to compute the embeddings.

## D    VARIANT WITH HANDCRAFTED FEATURES

We propose here to explore another variant of the PASTA-metric using handcrafted features instead of CLIP embeddings. The goal here is to avoid bias related to the use of such an embedding model by using an interpretable process to describe the extraction process.

### D.1    ADDITIONAL NOTATIONS

Let us now define additional notation that will be used for this section. First, we consider that we have a dataset $\mathcal{D}_l = \{\boldsymbol{x}_i, y_i\}_{i=1}^{N_l}$, where $l$ is the index of the training dat aset. $l \in [\![0, 4]\!]$ and $N_l$ is the number of data points in dataset $l$. On this dataset, we train two kinds of DNNs. First, we can train a simple DNN $f_\omega^{j_1}(\cdot)$ that outputs just a prediction $\hat{y}_i = f_\omega^{j_1}(\boldsymbol{x}_i)$, or we can train a greybox

Table 20: **Influence of the CBM explanation embedding process.** *Weightened* refers to the process described in Equation 14, Sentence refers to the process described in Equation 2, preceded with the template noted in *Template*. Each value is the average result on 5 runs with the standard deviation.

| *Computation* | *MSE* | *QWK* | *SCC* | *Template* |
|---|---|---|---|---|
| *Weighted* | $\mathbf{1.03 \pm 0.11}$ | $0.47 \pm 0.07$ | $0.24 \pm 0.25$ | – |
| *Sentence* | $1.06 \pm 0.05$ | $\mathbf{0.48 \pm 0.05}$ | $\mathbf{0.25 \pm 0.25}$ | " " |
| *Sentence* | $1.11 \pm 0.05$ | $0.47 \pm 0.05$ | $0.24 \pm 0.24$ | "The model's prediction is motivated by the concepts:" |

DNN that can predict both an explanation and a prediction $(\hat{y}_i, e_i^{j_3}) = f_\omega^{j_3}(\boldsymbol{x}_i)$. Here, $\omega$ represents the weights of the DNN.

For post-hoc methods we consider that we have a model $g^{j_2}(\cdot)$ that might have parameters or not and that we apply on $f_\omega^{j_1}(\cdot)$ to output an explanation such that $e_i^{j_1,j_2} = g^{j_2}(f_\omega^{j_1}(\boldsymbol{x}_i))$. The indices $j_1$ and $j_3$ account for the index of models, while $j_2$ is an index related to the number of post-hoc explanations. For clarity, we use $j$ as the index for the explanation techniques, which could be linked to $j_2$ and $j_1$, or only $j_3$. Let us consider that we have $j \in [\![0, 45]\!]$ kinds of explanations.

With these DNNs, we can now have the dataset $\mathcal{D}_l^{\textbf{XAI}} = \{e_i^j, m_i^j\}_{(i,j)\in[0,N_l']\times[0,N_J]}$, with $N_J = 45$ being the number of explanations and $N_l' = 24$ the number of test images for each dataset $l$. Here, $m_i^j$ represents an average mark provided by annotators that we aim to estimate. To achieve this, we propose a new model (that could be a DNN) $h_\omega(\cdot)$ that takes as input $e_i^j$ or a representation $\phi(e_i^j)$ of this explanation and outputs $\hat{m}_k^j = h_\omega(\phi(e_i^j))$ to approximate $m_i^j$. In the next section, we detail our architectural choice for $h_\omega(\cdot)$.

## D.2 NETWORK ARCHITECTURE

Before delving into the architecture, let us first describe the representation space. To ensure a uniform representation of all data, we have decided that $\phi(\cdot)$ should be a fixed-size vector where each coordinate represents a criterion $c_k$ that assesses a specific aspect of the XAI methods. This can be expressed as:

$$\phi(e_i^j) = [c_1 \quad \ldots \quad c_K]. \tag{15}$$

This technique includes three types of criteria, which we will describe in the following section.

## D.3 VARIANCE CRITERION

The goal of the Variance criterion is to assess the stability of an explanation. To do so, we propose to perform data augmentation on the images and measure the variance of the explanation, normalized by the mean of the explanation. First, let us define three sets of data augmentations: $\mathcal{A}^1$, $\mathcal{A}^2$, and $\mathcal{A}^3$. The resulting variance criterion is, for post hoc explanations:

$$c_k = \frac{\sum_{aug\in\mathcal{A}} \left[g^{j_2}(f_\omega^{j_1}(aug(\boldsymbol{x}_i)))\right]^2}{|\mathcal{A}|} - \frac{\left[\sum_{aug\in\mathcal{A}} \left[g^{j_2}(f_\omega^{j_1}(aug(\boldsymbol{x}_i)))\right]\right]^2}{|\mathcal{A}|^2}. \tag{16}$$

Or, for ante-hoc models:

$$c_k = \frac{\sum_{aug\in\mathcal{A}} \left[f_\omega^{j_3}(aug(\boldsymbol{x}_i))\right]^2}{|\mathcal{A}|} - \frac{\left[\sum_{aug\in\mathcal{A}} \left[f_\omega^{j_3}(aug(\boldsymbol{x}_i))\right]\right]^2}{|\mathcal{A}|^2}, \tag{17}$$

with $\mathcal{A} \in [\![\mathcal{A}^1, \mathcal{A}^2, \mathcal{A}^3]\!]$.

### D.4 COMPLEXITY CRITERION

The goal of the Complexity criterion is to evaluate the minimal dimensionality of the explanation. To achieve this, we use Principal Component Analysis (PCA) for each dataset $\mathcal{D}_l^{\mathbf{XAI}}$. Let $\lambda_p$ denote the eigenvalue associated with the $p$-th principal component. Our criterion is defined as:

$$c_k = \frac{\sum_{p \in [0,a]} \lambda_p}{\sum_p \lambda_p},$$ (18)

where $a$ is an integer.

### D.5 CLASSIFICATION CRITERION

The goal of the Classification criterion is to evaluate the model's capacity to discriminate between different classes based on the given explanation as a representation. The basic idea is that if a classifier is effective, then that means using the explanation as a representation explanation should provide sufficient information for each class to discriminate between them. Let $\mathrm{class}(\mathcal{D}_l^{\mathbf{XAI},j})$ denote a classifier model trained on $\mathcal{D}_l^{\mathbf{XAI},j}$ (the index $j$ means that we focus just on the $j$-th explanation), and let $\mathrm{accu}\left(\mathrm{class}(\mathcal{D}_l^{\mathbf{XAI}})\right)$ represent the accuracy of the classifier model trained and tested on $\mathcal{D}_l^{\mathbf{XAI},j}$. Classification Criterion is defined by :

$$c_k = \mathrm{accu}\left(\mathrm{class}(\mathcal{D}_l^{\mathbf{XAI},j})\right).$$ (19)

## E APPLICATIONS

### E.1 EXAMPLES

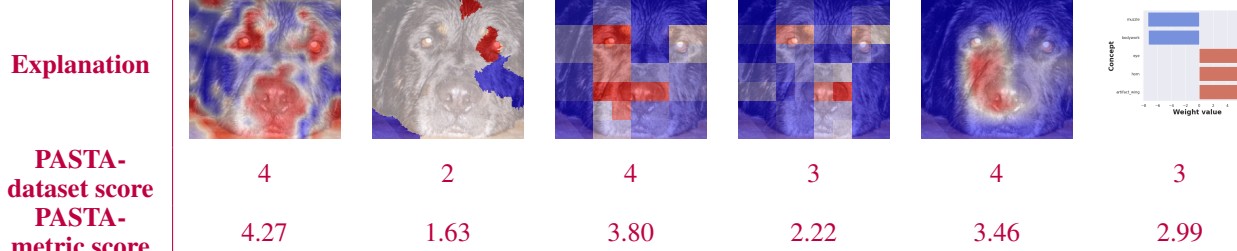

| | | | | | | |
|---|---|---|---|---|---|---|
| **Explanation** | | | | | | |
| **PASTA-dataset score** | 4 | 2 | 4 | 3 | 4 | 3 |
| **PASTA-metric score** | 4.27 | 1.63 | 3.80 | 2.22 | 3.46 | 2.99 |

Table 21: **Comparison of PASTA dataset and metric scores for given explanations.** These examples come from the use case of Q1. "PASTA-dataset score" indicates the mode among the 5 ratings given by annotators for this question. "PASTA-metric score" indicates the emulated score using the PASTA-metric.

### E.2 FINE TUNING OF XAI HYPERPARAMETERS

One use case of the PASTA-score is to fine-tune hyperparameters of XAI methods. We present a concrete case below: since applying LIME (Ribeiro et al., 2016) on images involves an optimizing process, there are many hyperparameters that can increase the quality of the explanation. However, knowing which parameters to use can be tricky and time-consuming because it involves judging manually the explanation generated for several parameters. Using PASTA-score, we can automate the process by searching the set of hyperparameters that maximize the PASTA-score. For example, one of the hyperparameters is the kernel width for the exponential kernel used to blur the images of the perturbating set. By doing so, we discovered that a kernel size of 0.15 (instead of the default

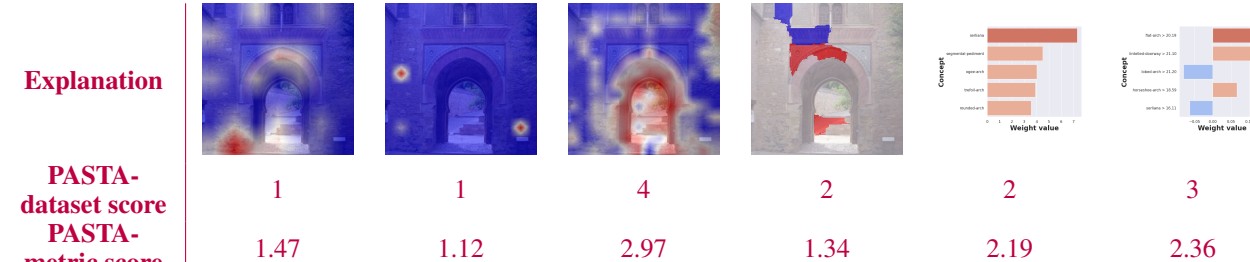

| | | | | | |
|---|---|---|---|---|---|
| **Explanation** | | | | | |
| **PASTA-dataset score** | 1 | 1 | 4 | 2 | 2 | 3 |
| **PASTA-metric score** | 1.47 | 1.12 | 2.97 | 1.34 | 2.19 | 2.36 |

Table 22: **Comparison of PASTA dataset and metric scores for given explanations.** These examples come from the use case of Q3. "PASTA-dataset score" indicates the mode among the 5 ratings given by annotators for this question. "PASTA-metric score" indicates the emulated score using the PASTA-metric.

value that is 0.25) increases the PASTA-score of LIME (ResNet50) from 2.16 to 2.25 (See Table 23), resulting in a slightly better explanations. We put examples of explanations produced by both hyperparameter settings in Figure 25.

| Kernel Width | PASTA-score |
|---|---|
| 0.05 | 2.15 |
| 0.1 | **2.25** |
| 0.15 | 2.18 |
| 0.2 | 2.13 |
| 0.25 | 2.16 |
| 0.3 | 2.17 |
| 0.35 | 2.17 |

Table 23: **Impact of Kernel Width on the PASTA-score**

| | |
|---|---|
| **Explanation with default parameters (Kernel Width = 0.25)** | 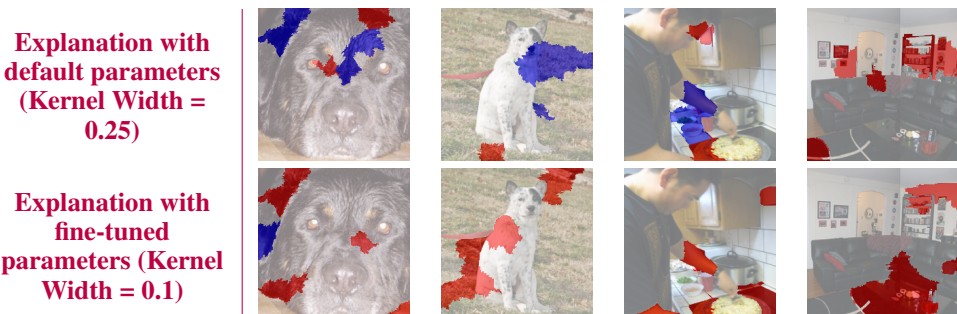 |
| **Explanation with fine-tuned parameters (Kernel Width = 0.1)** | |

Table 24: **Comparison of explanations obtained with default LIME parameters and fine-tuned parameters with PASTA-score.** Images tend to be slightly more focused on distinct objects, like the hands in the third image or the table in the fourth image. They also have less counterintuitive observations, like the negative contribution of pixels of the dog in the second image.

Another use case is the sample-level search of the target layer of a GradCAM explanation. Considering a fixed image. We computed the PASTA-score and looked for the best score. As a result, we observe that the highest scoring setup is the one that is closer to the end, respecting the observation that the deepest layers capture the higher-level features.

| Layer | relu_5_3 | relu_5_2 | relu_5_1 | relu_4_3 |
|---|---|---|---|---|
| Explanation | 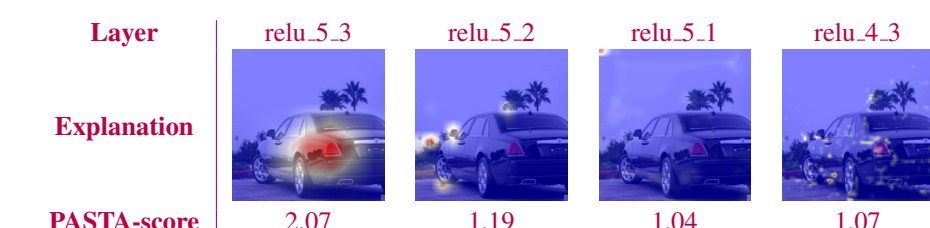 | | | |
| PASTA-score | 2.07 | 1.19 | 1.04 | 1.07 |

Table 25: **Comparison of different PASTA-scores for different layers selected in GradCAM.**

