# OpenReview forum: "Benchmarking XAI Explanations with Human-Aligned Evaluations"
_ICLR.cc/2025/Conference — Submitted to ICLR 2025_

### Official Review · Reviewer_1rMP · 2024-10-30

**Soundness:** 3
**Presentation:** 2
**Contribution:** 3
**Rating:** 5
**Confidence:** 4

**Summary:**

This paper proposes backbone for XAI, introducing a human evaluation protocol. The authors construct a benchmark dataset consisting of triplets of images, explanations, and labels, allowing for quantitative evaluation of XAI methods. Additionally, the paper consolidates different evaluation criteria and presents a question-based protocol for data annotation.

**Strengths:**

1. The paper conducts a comprehensive investigation of XAI methods, providing a convincing and promising set of evaluation criteria and protocols that offer a strong foundation for future research.

2. Experiments demonstrate that PASTA achieves human-level or better performance under the proposed new protocol.

**Weaknesses:**

1. The process of generating computed explanations and annotating concepts is unclear. The PASTA-metric’s applicability to unseen datasets or new XAI methods appears limited, as the generalization to other contexts or modalities is not fully validated. The four datasets used may not represent the full range of XAI applications, potentially limiting the framework’s relevance for domains beyond standard object recognition. From Table 5 in the supplementary material, it seems that concepts are defined by class names in the datasets. Would these pre-defined concepts limit the model’s generalization? If concepts are simply classes, it may keep the model as a “black-box”. So how to define the explainability?

2. Including more figures and equations to illustrate the metrics would be beneficial. For instance, how are scores for faithfulness calculated? This additional detail would enhance clarity.

3. Participant Background: The study included 15 participants—are they experts in this field? Is this number sufficient for robust annotation?

4. The main sections are somewhat unclear, and parts seem to serve as an overview of the supplementary material, which makes the paper challenging to follow. I recommend the authors provide a clearer narrative or at least one example before referring to supplementary sections.

5. There are minor typos, such as “tree losses” in Line 1563, which should be “three losses,” and issues with brackets in Fig. 8.

6. I also recommend comparing PASTA with traditional protocols or providing a visualization to demonstrate the effectiveness of the backbones.

**Questions:**

Please see the points under Weaknesses above.

---

> ### Author Response · Authors · 2024-11-18
> **Answer to reviewer 1rMP**
>
> We appreciate the reviewer’s comments and will address them in the following seven points:
>
> ### 1. Process of Generating Computed Explanations and Annotating Concepts (W1):
>    We thank the reviewer for raising this point and have revised the manuscript to provide a clearer explanation of how computed explanations are generated and concepts are annotated. Specifically, we have included details about the pipeline for generating explanations, the criteria for selecting annotators, and the methodology for annotating concepts. Additionally, we have added a visual illustration in the main paper to improve clarity and facilitate understanding.
>
> ### 2. Applicability of PASTA to Unseen Datasets or XAI Methods (W1):
>    We acknowledge the importance of validating PASTA's generalization to other contexts and modalities. As shown in Table 3, PASTA has been applied to unseen images from the datasets, demonstrating its adaptability. However, we emphasize that the primary goal of this work is to provide a benchmark that enables fair and consistent comparisons of XAI methods. While the diversity of datasets and methods in the current version is limited, PASTA represents a significant advancement over prior works by introducing a larger number of methods and a greater total number of samples. To offer a more comprehensive perspective, we have added a comparison with existing benchmarks in the appendix (Section: Comparison with Existing Benchmarks)
>
>    | Name               	| Annotations 	| N_Samples  | N_Part | Modality | N_Q | N_XAI | N_Data |
>    |----------------------------|---------------------|---------------|------------|--------------|----------|-----------|------------|
>    | PASTA-dataset          	| Likert         	| 66,000    	| 15     	| I + C    	| 6    	| 21    	| 100    	|
>    | Yang et al. (2022) [1]     	| Saliency, 2AFC 	| 356       	| 46     	| I        	| 2    	| 1     	| 89     	|
>    | Colin et al. (2022) [2]    	| Classification 	| 1,960     	| 241    	| I        	| 1    	| 6     	| NA     	|
>    | Dawoud et al. (2023) [3]   	| Clicktionary   	| 3,836     	| 76     	| I        	| 1    	| 3     	| 102    	|
>    | Mohseni et al. (2021) [4]  	| Saliency       	| 1,500     	| 200    	| I + T    	| 1    	| No    	| 1,500  	|
>    | Herm et al. (2021)[5]     	| Likert         	| NA        	| 165    	| C        	| 1    	| 6     	| NA     	|
>    | Morrison et al. (2023) [6] 	| Clicktionary/QCM   | 450       	| 50     	| I        	| 1    	| 3     	| 39     	|
>    | Spreitzer et al. (2022) [7]	| Likert/Binary  	| 4,050     	| 135    	| C        	| 9    	| 2     	| NA     	|
>    | Xuan et al. (2023) [8]    	| Likert/Binary  	| 3,600     	| 200    	| C        	| 4    	| 2     	| 1,326  	|
>
>    The table below provides an overview of datasets and human evaluation frameworks for XAI methods. Name specifies the reference of the dataset utilized in each study. Annotations describe the type of labels used in the dataset. N_{Samples} indicates the total number of samples that make up the dataset, while N_{Part} represents the number of participants involved in the labeling process. The column Modality identifies the types of data the dataset includes: I refers to images, C to concepts, and T to text. N_{Q} denotes the number of distinct questions posed to the annotators during the study. N_{XAI} refers to the number of XAI methods tested within the experiments, with No indicating cases where the dataset was used to label ground truth explanations without direct comparison to XAI methods. Finally, N_{Data} represents the number of unique data samples (e.g., images) shown to annotators during the experiments.
> Regarding the types of human annotations used in existing studies (as shown in the "Annotations" column of the table), Likert refers to the use of a Likert scale for scoring, Saliency refers to pseudo-saliency map evaluations, 2AFC indicates two-alternative forced choices, Clicktionary corresponds to the click-based annotation game defined in [4], MCQ represents multiple-choice questions, and Binary refers to binary decision tasks. Our novelty lies in the larger volume of annotations compared to previous approaches, the large number of XAI techniques, and the use of transparent, post-hoc methods.
> ### 3. Clarity on Metric Calculations and Visual Aids (W2):
>    To address the reviewer’s concerns, we have added new figures and equations in Section B.2 (Comparison with Existing Metrics), including formulas and visual aids to illustrate the process of computing our scores. These additions aim to improve the clarity and accessibility of the metric explanations.

---

> > ### Author Response · Authors · 2024-11-18
> > **Second part of the answer to reviewer 1rMP**
> >
> > ### 4. Participant Background and Annotation Robustness (W3):
> >    We understand the reviewer’s concerns regarding the number and expertise of participants. To ensure a broad perspective, our study included a diverse group of participants with varying levels of familiarity with XAI. All annotators underwent comprehensive training on XAI concepts, facilitated by a contracted company, with several hours of instruction. The questions asked to the annotators were validated by a psychologist and the annotation tool by a human-machine interaction expert. While we cannot claim the annotators are XAI experts, this training process ensures a foundational understanding of the concepts behind XAI. We will discuss this limitation in the revised manuscript.
> >
> > ### 5. Clarity of main sections and supplementary material (W4):
> >    We appreciate the reviewer’s suggestion to improve the narrative and provide examples. To address this, we will revise the manuscript by incorporating clearer explanations and detailed examples that illustrate key concepts. Specifically, we will add case studies or practical applications to demonstrate the relevance of the methods discussed.
> >
> > ### 6. Minor typos and figure issues (W5):
> >    Thank you for pointing out these issues. We will correct the typo and address the errors in Fig. 8 in the revised manuscript.
> >
> > ### 7. Comparison with Traditional Protocols and Backbone Visualization (W6):
> >    We appreciate the reviewer’s suggestion to enhance the comparison with traditional protocols and to include visualizations. In the revised manuscript, we have conducted additional studies to compare the PASTA-metric with existing alternatives. Specifically, we tested variations in the feature extraction process, such as replacing the CLIP encoder with the LLaVA encoder and using handcrafted feature extraction techniques. Additionally, we evaluated alternative scoring models, including Ridge Regression, Lasso Regression, Support Vector Machines, and a Multi-Layer Perceptron with a single hidden layer of 100 units. These experiments are now discussed in detail to provide a comprehensive comparison in sections 4.3 CLASSIFIER RESULTS and C.4 SCORING FUNCTIONS.
> >
> > Experiments about different extraction techniques:
> >
> > | Scoring Function | MSE  | QWK  | SCC  |
> > |------------------|------|------|------|
> > | PASTA        	| 1.06 | 0.48 | 0.25 |
> > | SVM          	| 0.97 | 0.39 | 0.22 |
> > | Ridge        	| 0.98 | 0.37 | 0.22 |
> > | Lasso        	| 1.71 | 0.31 | 0.16 |
> > | MLP          	| 1.28 | 0.38 | 0.20 |
> >
> > This table presents Mean Square Error (MSE), Quadratic Weighted Kappa (QWK), and Spearman Correlation Coefficient (SCC) metrics for different extraction techniques.
> >
> > Experiments about different scoring networks:
> >
> > | Metric   | Model                      	| Q1	| Q2	| Q3	| Q4	| Q5	| Q6	|
> > |----------|--------------------------------|-------|-------|-------|-------|-------|-------|
> > | MSE  	| PASTA-metric (CLIP)       	| 1.06  | 1.13  | 1.21  | 1.15  | 1.96  | 0.76  |
> > |      	| PASTA-metric (LLaVa)      	| 1.02  | 1.04  | 1.28  | 1.08  | 1.66  | 1.13  |
> > |      	| Feature Extraction        	| 4.50  | 5.81  | 3.71  | 3.61  | 3.54  | 3.58  |
> > |      	| Human                     	| 0.53  | 0.51  | 0.74  | 0.72  | 1.00  | 0.52  |
> > |----------|--------------------------------|-------|-------|-------|-------|-------|-------|
> > | QWK  	| PASTA-metric (CLIP)       	| 0.48  | 0.44  | 0.43  | 0.43  | 0.32  | 0.48  |
> > |      	| PASTA-metric (LLaVa)      	| 0.43  | 0.45  | 0.40  | 0.42  | 0.36  | 0.00  |
> > |      	| Feature Extraction        	| 0.09  | 0.09  | 0.03  | 0.03  | 0.05  | 0.02  |
> > |      	| Human                     	| 0.73  | 0.74  | 0.63  | 0.62  | 0.65  | 0.59  |
> > |----------|--------------------------------|-------|-------|-------|-------|-------|-------|
> > | SCC  	| PASTA-metric (CLIP)       	| 0.25  | 0.23  | 0.23  | 0.22  | 0.17  | 0.24  |
> > |      	| PASTA-metric (LLaVa)      	| 0.23  | 0.24  | 0.21  | 0.22  | 0.20  | 0.00  |
> > |      	| Feature Extraction        	| 0.16  | 0.09  | 0.14  | 0.22  | 0.17  | 0.11  |
> > |      	| Human                     	| 0.37  | 0.38  | 0.33  | 0.33  | 0.34  | 0.29  |
> >
> > This table presents Mean Square Error (MSE), Quadratic Weighted Kappa (QWK), and Spearman Correlation Coefficient (SCC) metrics for several strategies to compute scores (More information in Section C.1.2 EVALUATION METRICS). These results were obtained using CLIP and LLaVa [9] as multimodal encoders, referred to as $\text{PASTA-metric}^{\text{CLIP}}$ and $\text{PASTA-metric}^{\text{LLaVa}}$, respectively. We also tested an alternative approach involving handcrafted feature extraction, as described in Section D VARIANT WITH HANDCRAFTED FEATURES. Human refers to inter-annotator agreement metrics.

---

> > > ### Author Response · Authors · 2024-11-18
> > > **Third part of the answer to reviewer 1rMP**
> > >
> > > references
> > >
> > > [1] Scott Cheng-Hsin Yang, Nils Erik Tomas Folke, and Patrick Shafto. A psychological theory of explainability. In International Conference on Machine Learning, pp. 25007–25021. PMLR,
> > > 2022.
> > >
> > > [2] Julien Colin, Thomas Fel, R´emi Cad`ene, and Thomas Serre. What I cannot predict, I do not understand: A human-centered evaluation framework for explainability methods. Advances in NeuralInformation Processing Systems, 35:2832–2845, 2022.
> > >
> > > [3] Karam Dawoud, Wojciech Samek, Peter Eisert, Sebastian Lapuschkin, and Sebastian Bosse. Human-centered evaluation of XAI methods. In 2023 IEEE International Conference on Data
> > > Mining Workshops (ICDMW), pp. 912–921. IEEE, 2023.
> > >
> > > [4] Sina Mohseni, Jeremy E. Block, and Eric Ragan. Quantitative evaluation of machine learning explanations: A human-grounded benchmark. In 26th International Conference on Intelligent User Interfaces, pp. 22–31, 2021.
> > >
> > > [5] Lukas-Valentin Herm, Jonas Wanner, Franz Seubert, and Christian Janiesch. I don’t get it, but it seems valid! the connection between explainability and comprehensibility in (x) ai research. In ECIS, 2021
> > >
> > > [6] Katelyn Morrison, Mayank Jain, Jessica Hammer, and Adam Perer. Eye into ai: Evaluating the
> > > interpretability of explainable ai techniques through a game with a purpose. Proceedings of the ACM on Human-Computer Interaction, 7(CSCW2):1–22, 2023.
> > >
> > > [7] Nina Spreitzer, Hinda Haned, and Ilse van der Linden. Evaluating the practicality of counterfactual explanations. In XAI. it@ AI* IA, pp. 31–50, 2022.
> > >
> > > [8] Yueqing Xuan, Edward Small, Kacper Sokol, Danula Hettiachchi, and Mark Sanderson. Can users correctly interpret machine learning explanations and simultaneously identify their limitations? arXiv preprint arXiv:2309.08438, 2023
> > >
> > > [9] Liu, H., Li, C., Wu, Q., & Lee, Y. J. (2024). Visual instruction tuning. Advances in neural information processing systems, 36.

---

> > > > ### Author Response · Authors · 2024-11-23
> > > >
> > > > Dear Reviewer 1rMP,
> > > >
> > > > Thank you for taking the time to serve as a reviewer for our paper. We would like to kindly remind you that the rebuttal period will conclude in less than a week. As of now, we have not received any feedback from you. Could you please share your comments or suggestions with us?
> > > >
> > > > Best regards,

---

> > > > > ### Comment · Reviewer_1rMP · 2024-11-24
> > > > >
> > > > > First, I would like to thank the authors for addressing my requests in their response and for providing the revised version.
> > > > >
> > > > >
> > > > > Process of Generating Computed Explanations and Annotating Concepts (W1): Can you provide further detailed workflow for the generation of explanations, including the specific models or methods used, and how these interact with the annotators? Are these explanations purely model-generated, or is there a human-in-the-loop process involved?

---

> > > ### Comment · Reviewer_1rMP · 2024-11-24
> > >
> > > Participant Background and Annotation Robustness (W3): Can you provide justification for why 15 participants, considering the large dataset?  And how well these annotators were align with expert-level understanding of the domain are they XAI experts?
> > > Clarity of main sections and supplementary material (W4): As you mentioned Specifically, we will add case studies or practical applications to demonstrate the relevance of the methods discussed. Can you discuss one use case here
> > >
> > > Comparison with Traditional Protocols and Backbone Visualization (W6): What are the key takeaways from these comparisons?

---

> > > > ### Author Response · Authors · 2024-11-25
> > > >
> > > > We sincerely thank the reviewer for their thoughtful feedback and engagement with our work. It seems that W2 and W5 have already been addressed, so we will focus on the remaining points below.
> > > >
> > > > ### 1. Process of Generating Computed Explanations and Annotating Concepts (W1)
> > > > Thank you for requesting further details regarding the workflow for generating explanations and annotating concepts. Here is an outline of the process we followed:
> > > >
> > > > **Step 1**: Dataset Selection:
> > > > We chose four diverse datasets, COCO, PascalPART, CatsDogsCars and MonuMAI, to ensure broad applicability and generalizability.
> > > >
> > > > **Step 2**: Model Training:
> > > > For each dataset, we trained several DNNs using three backbones: ResNet-50, CLIP, and ViT. These were used for post-hoc explanation techniques. For XAI-by-design approaches, we adapted the backbones whenever feasible, ensuring a variety of methods and avoiding bias in explanation generation.
> > > >
> > > > **Step 3**: Test Set Construction:
> > > > We selected 25 images from the test set of each dataset to create the dataset specifically designed for evaluation and training of Pasta.
> > > >
> > > > **Step 4**: Explanation Extraction:
> > > > From the curated dataset, we generated 2200 explanations, ensuring representation across different methods and backbones.
> > > >
> > > > **Step 5**: Annotation Process:
> > > > These explanations were then reviewed by 15 trained annotators, who evaluated each explanation according to specific criteria.
> > > >
> > > > **Step 6**: Evaluation and Training:
> > > > The annotated dataset was split into training, validation, and test subsets. We used this dataset to train our proposed system, PASTA, and to evaluate the performance of different XAI techniques.
> > > >
> > > > Concerning specific models or methods used, the precise set of XAI methods and DNNs used are available in the table below:
> > > >
> > > > | Name                	| Applied on                      	|
> > > > |-------------------------|-------------------------------------|
> > > > | BCos                	| ResNet50-BCos                  	|
> > > > | GradCAM             	| ViT, ResNet50, CLIP (zero-shot)	|
> > > > | HiResCAM            	| ViT, ResNet50, CLIP (zero-shot)	|
> > > > | GradCAMElementWise  	| ViT, ResNet50, CLIP (zero-shot)	|
> > > > | GradCAM++           	| ViT, ResNet50, CLIP (zero-shot)	|
> > > > | XGradCAM            	| ViT, ResNet50, CLIP (zero-shot)	|
> > > > | AblationCAM         	| ViT, ResNet50, CLIP (zero-shot)	|
> > > > | ScoreCAM            	| ViT, ResNet50                  	|
> > > > | EigenCAM            	| ViT, ResNet50, CLIP (zero-shot)	|
> > > > | EigenGradCAM        	| ViT, ResNet50, CLIP (zero-shot)	|
> > > > | LayerCAM            	| ViT, ResNet50, CLIP (zero-shot)	|
> > > > | FullGrad            	| ViT, ResNet50                  	|
> > > > | Deep Feature Factorizations | ViT, ResNet50, CLIP (zero-shot) |
> > > > | SHAP                	| ViT, ResNet50, CLIP (zero-shot)	|
> > > > | LIME                	| ViT, ResNet50, CLIP (zero-shot)	|
> > > > | X-NeSyL             	| X-NeSyL                        	|
> > > > | CLIP-linear-sample  	| CLIP-linear                    	|
> > > > | CLIP-QDA-sample     	| CLIP-QDA                       	|
> > > > | LIME-CBM            	| CLIP-QDA, ConceptBottleneck    	|
> > > > | SHAP-CBM            	| CLIP-QDA, ConceptBottleneck    	|
> > > > | RISE                	| ConceptBottleneck              	|
> > > >
> > > > Name refers to the name of the XAI method, and Applied on refers to the different classifiers we computed the explanation on. For example we computed explanations using EigenGradCAM on ViT, ResNet50 and CLIP (zero-shot).
> > > >
> > > > It is important to note that the annotators were not involved in selecting the XAI techniques, ensuring independence in the evaluation process. They provided us with some feedback when they did not understand something but that is all.
> > > >
> > > > We hope this provides greater clarity on our methodology. Could you confirm to us that our explanation is clearer?

---

> > > > > ### Author Response · Authors · 2024-11-25
> > > > > **Second part**
> > > > >
> > > > > ### 2. Participant Background and Annotation Robustness (W3)
> > > > > 2.1 Participant Selection
> > > > > The choice of 15 annotators was a deliberate trade-off:
> > > > >
> > > > > - Diversity and Variability: Including a sufficient number of annotators allowed us to study individual variability and reduce variance in the mean scores.
> > > > > - Training Requirements: Each annotator underwent training to ensure they could accurately interpret and assess the explanations, which limited the feasible number of participants.
> > > > >
> > > > > Unlike many XAI benchmarks that rely on platforms like Amazon Mechanical Turk and use untrained participants, our annotators received dedicated training sessions. This approach ensured higher-quality annotations aligned with the study’s objectives.
> > > > >
> > > > > 2.2 Expertise of Annotators
> > > > > The annotators were not XAI experts by design. Instead, they represented end-users who would interact with XAI methods in real-world applications, such as healthcare professionals or non-technical stakeholders. Their training focused on enabling them to understand and interpret key concepts like saliency maps and concept weights. This approach reflects the intended use case of XAI methods for non-expert users. Their only specific training is made to ensure that they can interpret the explanations (e.g., what a saliency map represents, what is the positive or negative weight of a concept…) Therefore we have tried to mimic such realistic applications in the design of the evaluation protocol.
> > > > >
> > > > > 2.3 Dataset Scope and Redundancy
> > > > > We ensured robustness by having multiple annotators review the same data, enabling cross-validation of annotations and achieving inter-annotator reliability metrics. Annotation consistency was further validated and discussed in Section B.3.
> > > > >
> > > > > 2.4 Dataset Size
> > > > > Our dataset comprises 2200 explanations annotated by 15 evaluators, which compares favorably to benchmarks in similar domains (e.g., automated essay scoring datasets). We prioritized evaluating a broader range of XAI techniques over including more images, resulting in one of the largest XAI benchmarks published to date.
> > > > >
> > > > >
> > > > > ### 3. Clarity of Main Sections and Supplementary Material (W4)
> > > > > In response to W4, we have included additional experiments in **Appendix E.2** to demonstrate how our metrics can assist in hyperparameter tuning for XAI methods. Specifically:
> > > > > - **Experiment 1**: Using the PASTA-score to automate the selection of the optimal kernel width for the exponential kernel used in blurring images during the LIME perturbation set optimization process.
> > > > > - **Experiment 2**: Identifying among GradCAM explanations on different layer ResNet-50, the one with the highest PASTA-score and evaluating whether the resulting explanations align with expectations.
> > > > > Additional avenues for future research are outlined in the conclusion.
> > > > >
> > > > >
> > > > > ### 4. Comparison with Traditional Protocols and Backbone Visualization (W6)
> > > > >
> > > > > Regarding W6, we extracted several insights about design choices. Below, we summarize the main takeaways from our experiments:
> > > > > - **Comparison with Other Regression Processes:**
> > > > > Our results highlight interesting parallels between designing scoring networks for perceptual assessment and tasks such as essay scoring, despite their differences. Notably:
> > > > >   - Increasing the network's complexity, such as using a wider architecture like an MLP, does not necessarily improve performance.
> > > > >   - Instead, employing losses that penalize ranking discrepancies [1] and integrating cosine similarity measures [2] significantly enhance both SCC and QWK.
> > > > > - **Comparison with Alternative Embedding Processes:**
> > > > >   - The choice of backbone between CLIP and LLaVa has minimal impact, suggesting that similar levels of information are extracted regardless of the backbone architecture.
> > > > >   - However, the use of handcrafted features results in a notable decrease in SCC, QWK, and MSE. This aligns with observations from related fields such as image quality assessment [3], where deep neural networks have proven more effective in capturing perceptual aspects than handcrafted features, albeit at the expense of interpretability.
> > > > >
> > > > > We hope these clarifications address your concerns and provide additional insights into our findings. Please let us know if further elaboration is required.
> > > > >
> > > > >
> > > > > [1] Ruosong Yang, Jiannong Cao, Zhiyuan Wen, Youzheng Wu, and Xiaodong He. Enhancing automated essay scoring performance via fine-tuning pre-trained language models with combination of regression and ranking. In Findings of the Association for Computational Linguistics: EMNLP 2020, pp. 1560–1569, 2020
> > > > >
> > > > > [2] Yongjie Wang, Chuan Wang, Ruobing Li, and Hui Lin. On the use of BERT for automated essay scoring: Joint learning of multi-scale essay representation. arXiv preprint arXiv:2205.03835,2022.
> > > > >
> > > > > [3] Richard Zhang, Phillip Isola, Alexei A Efros, Eli Shechtman, and Oliver Wang. The unreasonable effectiveness of deep features as a perceptual metric. In Proceedings of the IEEE conference on computer vision and pattern recognition, pp. 586–595, 2018.

---

### Official Review · Reviewer_2wcf · 2024-10-30

**Soundness:** 3
**Presentation:** 2
**Contribution:** 2
**Rating:** 6
**Confidence:** 4

**Summary:**

This paper conducts a user study on the comparative effectiveness of different explainable AI (XAI) methods, and proposes an automatic XAI evaluation model. In particular, it asks human participants to rate different aspects of the explanations given carefully designed questions, and train a model that takes in the explanations (either text or saliency maps) to predict the ratings. Experimental results reveal the advantages of visual explanations over concept ones, and show the feasibility of predicting human ratings based on model explanations.

**Strengths:**

(1) This paper tackles an important challenge in XAI: the alignment between model explanations and human evaluations.

(2) It carries out human study with a broad range of explanation methods and multiple datasets, which can facilitate the development of future XAI methods.

(3) With an automatic model for estimating human ratings, the study can be extended to help improve the trustworthiness of deep networks.

(4) Extensive experiments are provided in the supplementary materials, providing the opportunity for more in-depth analyses.

**Weaknesses:**

(1) The paper emphasizes the limitations of existing studies in scaling to broader domains, due to difficulties of data collection and the subjectivity of human evaluation. Nevertheless, these problems are also well addressed in the proposed method. Specifically, the paper centers around a user study for rating model explanations, which also requires significant amounts of manual labor and does not alleviate annotator biases. While a quantitative evaluation model is proposed, with only the results in Table 3 (comparison to the baseline without explanations and inter-subject agreement), it is unclear how it can help evaluate XAI approaches in different domains. I would suggest involving the PASTA model for training other models, and exploring its effectiveness in enhancing the trustworthiness of deep networks.

(2) Despite the wide coverage of the user study, the analyses in the main paper are relatively shallow and fall short of providing important insights. In particular, while the paper highlights comparing over 10 classifiers and XAI methods, its conclusions focus on only two categories of XAI methods and three backbones. Table 1 indicates close-to-zero correlations between human ratings and the XAI methods, but without detailed explanations. It is reasonable to move certain results from the supplementary materials and perform more in-depth analyses following previous studies (e.g., [ref1, ref2]).

(3) The proposed PASTA model essentially feeds the visual or textual explanations to CLIP encoders to derive the human ratings. Such a design can have several limitations: First, CLIP is heavily tuned toward semantics and can have its own biases. It would be good to test with various designs, e.g., with recent approaches of projecting multi-modal data to a pretrained LLM space (e.g., [ref3]). Second, the model seems to only utilize explanations (or explanations on top of images) as inputs. While this is okay with simple visual stimuli that have a dominant object for classification, it may not generalize to other applications that demand complex reasoning with a rich set of visual elements (e.g., visual reasoning). It can be useful to consider at least including the ground truth answers as inputs. Third, relating to Table 3, I would expect comparisons with various methods, as the problem itself is just a regression.

References:

[ref1] Towards Human-Centered Explainable AI: A Survey of User Studies for Model Explanations. IEEE TPAMI, 2024.

[ref2] What I Cannot Predict, I Do Not Understand: A Human-Centered Evaluation Framework for Explainability Methods. NeurIPS, 2022.

[ref3] Visual Instruction Tuning. NeurIPS, 2023.

**Questions:**

(1) Please justify the difference between the proposed study and previous user studies on XAI.

(2) How can the PASTA model scale to broader domains, and help model development?

(3) Please consider reorganizing the paper, and including more in-depth analyses of the main paper

(4) It would be reasonable to perform a more comprehensive study on the automatic rating part.

---

> ### Author Response · Authors · 2024-11-18
> **Answer to reviewer 2wcf**
>
> We sincerely thank the reviewer for their comments and questions, which will help us strengthen the paper.
>
> ### 1. Methodological Concerns and Scope + model development (W1) + Q2
>
> We appreciate the reviewer’s insightful feedback and agree that the ultimate goal of our work is to enhance the quality of future XAI systems. To clarify, the primary aim of our work is to propose a perceptual criterion for XAI evaluation. This criterion allows researchers to assess the quality of their XAI techniques within our benchmark, making it easier to determine if an algorithm is effective. Our approach is inspired by advancements in the image quality assessment community. For many years, this field relied solely on human annotators to evaluate image quality until methods like LPIPS [1] introduced a standardized way to perform such assessments.
> While we agree with the reviewer that our criterion could also be used to improve XAI techniques, we chose to leave this application for future work due to the extensive scope of this paper. There are potential challenges in using the same criterion for both improvement and evaluation—if we refine a technique based on this criterion, how do we objectively assess it afterward?
>
> The PASTA benchmark and dataset currently deals with saliency-based, counterfactual-based, and concept-based explanation methods. Since the human evaluation protocol is normalized and has been developed with the help of a psychologist, it is reproducible and can be extended to new modalities with new annotators. This is why we intend to open source the data and the protocol is carefully described.
>
> We acknowledge the reviewer’s suggestion to elaborate further on the experiments of PASTA. In the revised version of the paper, we plan to expand this section and include the following table (currently found in 3.) to provide additional context.
>
> ### 2. Depth Analysis of the results and automated rating part (W2)
>
> 2.1: It is true that the results of the benchmark of Section 3.5 are relatively short. We have added some more comments in this Section. However, due to space limitations, extensive comments on the results of the benchmark are provided in Appendices B.2 and B.3. Concretely we discussed more the interpretations of why saliency-based methods seem to perform better, and a note about the higher performances of class independent explanations like EigenCAM.
>
> 2.2: Concerning the close-to-zero correlations between human ratings and existing XAI metrics, our interpretation, although quite short, is that human scores cover an aspect of explanation quality unrelated to that of perceptual quality, as previously noted by Biessmann & Refiano (2021). Indeed, on the one hand, human evaluations are most likely to accurately measure the usefulness of an explanation, which is ultimately targeted at a human audience. On the other hand, other computational metrics, such as faithfulness, are more likely to measure the adequation between the explanation and the real functioning of the model. This aspect is inherently inaccessible to a human judgment, as the human cannot access the internal functioning of the model. We have added a clarification on this in Section 3.6.

---

> > ### Author Response · Authors · 2024-11-18
> > **Second part of the answer to reviewer 2wcf**
> >
> > ### 3. Model Design, Experiments and Limitations (W3 + Q4)
> >
> > 3.1 The PASTA metric indeed relies on CLIP. To mitigate the dependence, we have added two new experiments:
> > - a PASTA metric similar to the CLIP-based one, where CLIP is replaced by LLaVa as a text-image encoder. The results are shown in Section 4.3 CLASSIFIER RESULTS.
> > - an alternative PASTA metric that does not depend on a neural network to align image and text embeddings. It is a low dimensional embedding based on several scores which can be computed both on saliency maps and concept importances. The details of these metrics have been added as Appendix D VARIANT WITH HANDCRAFTED FEATURES, and the results are shown in Section 4.3 CLASSIFIER RESULTS. However, it should be noted that this alternative metric is more computationally expensive, as some scores (e.g. Classification Criterion or Variance Criterion) require a lot of computations.
> >
> > 3.2 The prediction of the scores indeed only relies on the explanation, and does not take into account, e.g. the ground truth labels. We have added an additional experiment where we test this, presented in Section/Appendix C.3 ADD OF LABEL INFORMATION IN THE PASTA-METRIC EMBEDDING. The results are lower than without label information. We believe that this is because the number of labels used across all datasets (26) is comparable to the number of distinct images (100), inducing redundancy or overfitting.
> >
> > 3.3 Concerning the results of Table 3. We added in Section C.4 SCORING FUNCTIONS the results for Ridge Regression, Lasso Regression, Support Vector Machines, and a Multi-Layer Perceptron with a single hidden layer of 100 units, hence presenting 5 different models. The results of the different models are very close to each other, although the method presented in PASTA performs slightly better.
> >
> > | Metric 	| Model              	| Q1 | Q2 | Q3 | Q4 | Q5 | Q6 |
> > |----------------|----------------------------|--------|--------|--------|--------|--------|--------|
> > | MSE        	| PASTA-metric (CLIP)   	| 1.06   | 1.13   | 1.21   | 1.15   | 1.96   | 0.76   |
> > |            	| PASTA-metric (LLaVa)  	| 1.02   | 1.04   | 1.28   | 1.08   | 1.66   | 1.13   |
> > |            	| Feature Extraction    	| 4.50   | 5.81   | 3.71   | 3.61   | 3.54   | 3.58   |
> > |            	| Human                 	| 0.53   | 0.51   | 0.74   | 0.72   | 1.00   | 0.52   |
> > |----------------|----------------------------|--------|--------|--------|--------|--------|--------|
> > | QWK        	| PASTA-metric (CLIP)   	| 0.48   | 0.44   | 0.43   | 0.43   | 0.32   | 0.48   |
> > |            	| PASTA-metric (LLaVa)  	| 0.43   | 0.45   | 0.40   | 0.42   | 0.36   | 0.00   |
> > |            	| Feature Extraction    	| 0.09   | 0.09   | 0.03   | 0.03   | 0.05   | 0.02   |
> > |            	| Human                 	| 0.73   | 0.74   | 0.63   | 0.62   | 0.65   | 0.59   |
> > |----------------|----------------------------|--------|--------|--------|--------|--------|--------|
> > | SCC        	| PASTA-metric (CLIP)   	| 0.25   | 0.23   | 0.23   | 0.22   | 0.17   | 0.24   |
> > |            	| PASTA-metric (LLaVa)  	| 0.23   | 0.24   | 0.21   | 0.22   | 0.20   | 0.00   |
> > |            	| Feature Extraction    	| 0.16   | 0.09   | 0.14   | 0.22   | 0.17   | 0.11   |
> > |            	| Human                 	| 0.37   | 0.38   | 0.33   | 0.33   | 0.34   | 0.29   |
> >
> > This table presents Mean Square Error (MSE), Quadratic Weighted Kappa (QWK), and Spearman Correlation Coefficient (SCC) metrics for several strategies to compute scores (More information in Section C.1.2 EVALUATION METRICS). obtained using CLIP and LLaVa [10] as multimodal encoders, referred to as $\text{PASTA-metric}^{\text{CLIP}}$ and $\text{PASTA-metric}^{\text{LLaVa}}$, respectively. We also tested an alternative approach involving handcrafted feature extraction, as described in Section D VARIANT WITH HANDCRAFTED FEATURES. Human refers to inter-annotator agreement metrics.
> >
> > ### 4. Structure and Presentation (Q3)
> >
> > Inspired by comments from you and the other reviewers, we added more analyses of our results. Concretely, in Section 3.5 HUMAN EVALUATION AND RESULTS we discussed more the interpretations of why saliency-based methods seem to perform better, and a note about the higher performances of class independent explanations like EigenCAM. In section 3.6 CORRELATION WITH OTHER METRICS, we reasoned the results about uncorrelation between human assessments and computational metrics. Please refer to the general response for a comprehensive overview of our completed and planned revisions.

---

> > > ### Author Response · Authors · 2024-11-18
> > > **Third part of the answer to reviewer 2wcf**
> > >
> > > ### 5. User Studies related works (Q1)
> > >
> > > As far as our knowledge, we are the first study combining explanations based on saliency map and Concepts Bottleneck models. Secondly, the scope of our method, in terms of XAI methods evaluated and the number of annotated samples, is similar or higher compared to the existing dataset evaluating XAI methods. To give a better overview of existing benchmarks, we added a comparative comparison with the state of the art in the appendix (Section Comparison with existing benchmarks).
> > >
> > > | Name               	| Annotations 	| N_Samples  | N_Part | Modality | N_Q | N_XAI | N_Data |
> > > |----------------------------|---------------------|---------------|------------|--------------|----------|-----------|------------|
> > > | PASTA-dataset          	| Likert         	| 66,000    	| 15     	| I + C    	| 6    	| 21    	| 100    	|
> > > | Yang et al. (2022) [2]     	| Saliency, 2AFC 	| 356       	| 46     	| I        	| 2    	| 1     	| 89     	|
> > > | Colin et al. (2022) [3]    	| Classification 	| 1,960     	| 241    	| I        	| 1    	| 6     	| NA     	|
> > > | Dawoud et al. (2023) [4]   	| Clicktionary   	| 3,836     	| 76     	| I        	| 1    	| 3     	| 102    	|
> > > | Mohseni et al. (2021) [5]  	| Saliency       	| 1,500     	| 200    	| I + T    	| 1    	| No    	| 1,500  	|
> > > | Herm et al. (2021)[6]     	| Likert         	| NA        	| 165    	| C        	| 1    	| 6     	| NA     	|
> > > | Morrison et al. (2023) [7] 	| Clicktionary/QCM   | 450       	| 50     	| I        	| 1    	| 3     	| 39     	|
> > > | Spreitzer et al. (2022) [8]	| Likert/Binary  	| 4,050     	| 135    	| C        	| 9    	| 2     	| NA     	|
> > > | Xuan et al. (2023) [9]    	| Likert/Binary  	| 3,600     	| 200    	| C        	| 4    	| 2     	| 1,326  	|
> > >
> > > The table below provides an overview of datasets and human evaluation frameworks for XAI methods. Name specifies the reference of the dataset utilized in each study. Annotations describe the type of labels used in the dataset. N_{Samples} indicates the total number of samples that make up the dataset, while N_{Part} represents the number of participants involved in the labeling process. The column Modality identifies the types of data the dataset includes: I refers to images, C to concepts, and T to text. N_{Q} denotes the number of distinct questions posed to the annotators during the study. N_{XAI} refers to the number of XAI methods tested within the experiments, with No indicating cases where the dataset was used to label ground truth explanations without direct comparison to XAI methods. Finally, N_{Data} represents the number of unique data samples (e.g., images) shown to annotators during the experiments.
> > > Regarding the types of human annotations used in existing studies (as shown in the "Annotations" column of the table), Likert refers to the use of a Likert scale for scoring, Saliency refers to pseudo-saliency map evaluations, 2AFC indicates two-alternative forced choices, Clicktionary corresponds to the click-based annotation game defined in [4], MCQ represents multiple-choice questions, and Binary refers to binary decision tasks. Our novelty lies in the larger volume of annotations compared to previous approaches, the large number of XAI techniques, and the use of transparent, post-hoc methods.

---

> > > > ### Author Response · Authors · 2024-11-18
> > > > **Fourth part of the answer to reviewer 2wcf**
> > > >
> > > > references
> > > >
> > > > [1] Zhang, R., Isola, P., Efros, A. A., Shechtman, E., & Wang, O. (2018). The unreasonable effectiveness of deep features as a perceptual metric. In Proceedings of the IEEE conference on computer vision and pattern recognition (pp. 586-595).
> > > > [2] Scott Cheng-Hsin Yang, Nils Erik Tomas Folke, and Patrick Shafto. A psychological theory of explainability. In International Conference on Machine Learning, pp. 25007–25021. PMLR, 2022.
> > > > [3] Julien Colin, Thomas Fel, R´emi Cadene, and Thomas Serre. What I cannot predict, I do not understand: A human-centered evaluation framework for explainability methods. Advances in NeuralInformation Processing Systems, 35:2832–2845, 2022.
> > > > [4] Karam Dawoud, Wojciech Samek, Peter Eisert, Sebastian Lapuschkin, and Sebastian Bosse. Human-centered evaluation of XAI methods. In 2023 IEEE International Conference on Data Mining Workshops (ICDMW), pp. 912–921. IEEE, 2023.
> > > > [5] Sina Mohseni, Jeremy E. Block, and Eric Ragan. Quantitative evaluation of machine learning explanations: A human-grounded benchmark. In 26th International Conference on Intelligent User Interfaces, pp. 22–31, 2021.
> > > > [6] Lukas-Valentin Herm, Jonas Wanner, Franz Seubert, and Christian Janiesch. I don’t get it, but it seems valid! the connection between explainability and comprehensibility in (x) ai research. In ECIS, 2021
> > > > [7] Katelyn Morrison, Mayank Jain, Jessica Hammer, and Adam Perer. Eye into ai: Evaluating the interpretability of explainable ai techniques through a game with a purpose. Proceedings of the ACM on Human-Computer Interaction, 7(CSCW2):1–22, 2023.
> > > > [8] Nina Spreitzer, Hinda Haned, and Ilse van der Linden. Evaluating the practicality of counterfactual explanations. In XAI. it@ AI* IA, pp. 31–50, 2022.
> > > > [9] Yueqing Xuan, Edward Small, Kacper Sokol, Danula Hettiachchi, and Mark Sanderson. Can users correctly interpret machine learning explanations and simultaneously identify their limitations? arXiv preprint arXiv:2309.08438, 2023
> > > > [10] Liu, H., Li, C., Wu, Q., & Lee, Y. J. (2024). Visual instruction tuning. Advances in neural information processing systems, 36.

---

> > > > > ### Author Response · Authors · 2024-11-23
> > > > >
> > > > > Dear Reviewer 2wcf,
> > > > >
> > > > > Thank you for taking the time to serve as a reviewer for our paper. We would like to kindly remind you that the rebuttal period will conclude in less than a week. As of now, we have not received any feedback from you. Could you please share your comments or suggestions with us?
> > > > >
> > > > > Best regards,

---

> > > > > > ### Comment · Reviewer_2wcf · 2024-11-24
> > > > > >
> > > > > > I thank the authors for providing the clarification and additional results. The rebuttal addresses most of my questions regarding model design, additional analyses, and presentation. I do share a common concern with reviewer cj8g about the limited size of the dataset (i.e., whether 100 images are sufficient to draw generalizable conclusions), and I believe that experiments on leveraging the proposed metrics for model development would be an important extension for the study. I am leaning toward raising the score, and will make my final decision upon discussion with the other reviewers.

---

> > > > > > > ### Author Response · Authors · 2024-11-25
> > > > > > >
> > > > > > > We would like to thank you for your constructive feedback and take this opportunity to address your concerns with additional details and clarifications.
> > > > > > >
> > > > > > > ### 1. Dataset Size and Generalization Concerns
> > > > > > >
> > > > > > > We appreciate your concern regarding the dataset size (100 images). While the test set may seem limited in terms of images, it is essential to consider that each image is associated with a diverse set of explanations, resulting in a significantly larger number of data points. For example, our dataset contains **2200 annotated explanations** generated from these images, evaluated across multiple XAI techniques and backbones. In addition, the number of different images is comparable to other XAI benchmarks of this kind (see Table 8). The primary goal of the benchmark and metric is not to generalize to new images, but rather to new XAI metrics, as explained in Section 4.4. Furthermore, the human evaluation protocol has been designed in a modular and reproducible way with the help of a psychologist. This allows for including new images and/or new human evaluators in a potential future evaluation campaign. The open source release of the dataset and metrics is an important element to ensure that this work will widely benefit the XAI research community.
> > > > > > >
> > > > > > > ### 2. Proposed Metrics for Model Development
> > > > > > >
> > > > > > > Regarding the potential of leveraging the proposed metrics for model development, we want to clarify that this is not the main focus of the paper. We believe that overloading the manuscript with additional information might risk confusing the reader. However, acknowledging the reviewer’s perspective, we have included **additional experiments in Appendix E.2** to illustrate how our metrics can assist in hyperparameter tuning for XAI algorithms.
> > > > > > >
> > > > > > > Specifically, while we did not use the PASTA-score as a loss function for backpropagation, we employed it to identify the optimal hyperparameters for an XAI algorithm.
> > > > > > >
> > > > > > > - In **Tables 23 and 24**, we demonstrate how the PASTA-score can automate the selection of the best kernel width for the exponential kernel used in blurring images in the LIME perturbation set during the optimization process.
> > > > > > > - In **Table 25**, we apply the PASTA-score to analyze GradCAM explanations on different ResNet-50 layers, identifying the layer with the highest PASTA-score and evaluating whether the resulting explanations align with expectations.
> > > > > > >
> > > > > > > Additional research directions are discussed in the conclusion.

---

### Official Review · Reviewer_cta3 · 2024-11-09

**Soundness:** 2
**Presentation:** 3
**Contribution:** 1
**Rating:** 5
**Confidence:** 2

**Summary:**

This paper introduces a framework for human centric method of the explainable AI techniques (XAI). This paper performs evaluation on four datasets: COCO, Pascal Parts, Cats Dogs Cars and Monum AI.

**Strengths:**

1. The paper tries to answer an important question about the evaluation of the XAI methods.
2. The overall evaluation protocol and the research questions designed based on the fidelity, complexity, objectivity and robustness are presented in a right manner.

**Weaknesses:**

1. First and foremost to perform any sort of evaluation the authors should consider the methodology used and the architecture used specially in gradient based methods. eg: GradCAM does not work well with transformers, rather it is a methodology that works significantly better on CNNs due to the architectural composition. Chefer et. al. Transformer interpretability beyond attention visualization.
2. One of the major limitation of this work is use of another network suck as CLIP as explanations evaluator, this limits the model to the limitations of the CLIP. eg: CLIP embedding space limits it for the fine grained understanding in the joint embedding space.
3. The paper is more suitable for a user-based study conference rather than ICLR.

**Questions:**

Please check the limitations and kindly answer the correlation of the design of XAI methods and a deep learning architecture for the evaluation of the XAI methods. Another question that needs to be highlighted are the use of CLIP in the pipeline as it will be dependent on the embedded representations on the CLIP space.

---

> ### Author Response · Authors · 2024-11-18
> **Answer to reviewer cta3**
>
> We would like to thank reviewer cta3 for his or her questions on the adequation between the deep learning model and the XAI methods, and on the dependence of our new metric on CLIP.
>
> ---
>
> ### 1. Methodology and Architecture Dependency (W1)
>
> It is true that many XAI methods were originally designed for specific deep convolutional neural network architectures and may produce suboptimal results when applied to different models. However, we believe it is essential to evaluate how various XAI techniques perform across multiple backbones to ensure a fairer and more comprehensive assessment. This is precisely why we tested saliency-based methods (such as GradCAM and its variants) on different models, referred to as “backbones” in the paper. To this end, we collected human evaluations of 47 XAI methods across specific backbones, though we focused on 21 distinct XAI techniques in our analysis. For instance, GradCAM was tested on ResNet50, ViT-B, and CLIP zero-shot (which is based on ViT but uses a different training process). Similarly, SHAP-CBM was evaluated with both CLIP-QDA and the original Concept Bottleneck architecture proposed by Koh et al. [1]. We have added a paragraph clarifying this point in Section 3.2. Moreover, the results in Section 3.5 highlight a potential bias in the design of certain XAI methods, particularly favoring ResNet50.
>
> ---
>
> ### 2. Limitations of Using CLIP (W2)
>
> First, we would like to insist on the fact that CLIP is only used as a tool to build the PASTA metric, which extends the results of the benchmark to potential new XAI methods. The PASTA benchmark itself is independent of CLIP. For the PASTA metric, it indeed relies on CLIP. To mitigate the dependence, we have added two new experiments:
> - A PASTA metric similar to the CLIP-based one, where CLIP is replaced by LLaVa as a text-image encoder. The results are shown in Section 4.3.
> - An alternative PASTA metric that does not depend on a neural network to align image and text embeddings. It is a low-dimensional embedding based on several scores that can be computed both on saliency maps and concept importances. The details of these metrics have been added as Appendix D, and the results are shown in Section 4.3.
>
> However, it should be noted that this alternative metric is more computationally expensive, as some scores (e.g., Classification Criterion or Variance Criterion) require a lot of computations.
>
> | Metric  	| Model              	| Q1  | Q2  | Q3  | Q4  | Q5  | Q6  |
> |-------------|------------------------|------|------|------|------|------|------|
> | MSE     	| PASTA-metric (CLIP)   | 1.06 | 1.13 | 1.21 | 1.15 | 1.96 | 0.76 |
> |         	| PASTA-metric (LLaVa)  | 1.02 | 1.04 | 1.28 | 1.08 | 1.66 | 1.13 |
> |         	| Feature Extraction	| 4.50 | 5.81 | 3.71 | 3.61 | 3.54 | 3.58 |
> |         	| Human             	| 0.53 | 0.51 | 0.74 | 0.72 | 1.00 | 0.52 |
> | QWK     	| PASTA-metric (CLIP)   | 0.48 | 0.44 | 0.43 | 0.43 | 0.32 | 0.48 |
> |         	| PASTA-metric (LLaVa)  | 0.43 | 0.45 | 0.40 | 0.42 | 0.36 | 0.00 |
> |         	| Feature Extraction	| 0.09 | 0.09 | 0.03 | 0.03 | 0.05 | 0.02 |
> |         	| Human             	| 0.73 | 0.74 | 0.63 | 0.62 | 0.65 | 0.59 |
> | SCC     	| PASTA-metric (CLIP)   | 0.25 | 0.23 | 0.23 | 0.22 | 0.17 | 0.24 |
> |         	| PASTA-metric (LLaVa)  | 0.23 | 0.24 | 0.21 | 0.22 | 0.20 | 0.00 |
> |         	| Feature Extraction	| 0.16 | 0.09 | 0.14 | 0.22 | 0.17 | 0.11 |
> |         	| Human             	| 0.37 | 0.38 | 0.33 | 0.33 | 0.34 | 0.29 |
>
> This table presents Mean Square Error (MSE), Quadratic Weighted Kappa (QWK), and Spearman Correlation Coefficient (SCC) metrics for several strategies to compute scores obtained using CLIP and LLaVa [2] as multimodal encoders, referred to as $\text{PASTA-metric}^{\text{CLIP}}$ and $\text{PASTA-metric}^{\text{LLaVa}}$, respectively. We also tested an alternative approach involving handcrafted feature extraction, as described in Appendix D. Human refers to inter-annotator agreement metrics.

---

> ### Author Response · Authors · 2024-11-18
> **Second part of the answer to reviewer cta3**
>
> ### 3. Suitability for Conference Venue (W3)
>
> We respectfully disagree with the reviewer’s comment. Our manuscript directly addresses topics explicitly listed in the ICLR 2025 Call for Papers, namely “datasets and benchmarks” (as we propose a benchmark and release a dataset) and “societal considerations” (since explainability methods and their human evaluation contribute to enhancing the safety and fairness of machine learning systems). For these reasons, we believe our paper is well-suited for the conference. Additionally, this is not a user study for two key reasons: (1) annotators are not users, as they are not interacting with or using a system, and (2) the objective is not to interpret human annotations but rather to assess the performance of XAI techniques.
>
> ---
>
> ### References
>
> [1] Koh, Pang Wei, et al. "Concept bottleneck models." International conference on machine learning. PMLR, 2020.
>
> [2] Liu, H., Li, C., Wu, Q., & Lee, Y. J. (2024). Visual instruction tuning. Advances in neural information processing systems, 36.

---

> > ### Author Response · Authors · 2024-11-23
> >
> > Dear Reviewer cta3,
> >
> > Thank you for taking the time to serve as a reviewer for our paper. We would like to kindly remind you that the rebuttal period will conclude in less than a week. As of now, we have not received any feedback from you. Could you please share your comments or suggestions with us?
> >
> > Best regards,

---

> ### Comment · Reviewer_cta3 · 2024-11-25
>
> First and foremost, I would like to thank the authors for the reply on the reviews. One of my major concerns for this method is the dependency on the CLIP model (W2). The authors have replaced it with LLaVa, the problem is it will still be dependent on the internal architecture of the model, in this case LLaVa. The response to W1 is not satisfactory. But I appreciate the authors for the additional results they put in the paper. I would tend to increase my score from 3 to 5.

---

> > ### Author Response · Authors · 2024-11-25
> >
> > We sincerely thank you for your thoughtful engagement with our work and for your decision to increase the score. Below, we address the points you raised:
> >
> > ### Concerning W2 (Dependency on the CLIP Model)
> > In the revised manuscript, we proposed two alternative options to address the dependency on the CLIP model:
> >
> > 1. **Replacement with LLaVa**:
> >    We replaced CLIP with LLaVa, demonstrating that the choice of joint text and image embedding model has a limited influence on the results. This indicates that our methodology is not overly reliant on any specific model architecture.
> >
> > 2. **Handcrafted Features**:
> >    To further reduce dependency on external models, we introduced a second option based on handcrafted features. This approach uses an explicit joint embedding that does not rely on any external embedding model or internal architecture. Details of this computation are provided in Appendix D, and the comparative results are discussed in Section 4.3.
> >
> > We hope these additions address your concerns regarding model dependency. We would be happy to perform any experiment proposed by the reviewer.
> >
> > ### Concerning W1
> > We appreciate your feedback on W1 and would be grateful if you could provide further details on your remaining concerns. This would help us provide additional clarifications or results to address them comprehensively.
> >
> > Once again, we thank you for your constructive feedback and are committed to improving the clarity and rigor of our work.

---

### Official Review · Reviewer_cj8g · 2024-11-09

**Soundness:** 3
**Presentation:** 3
**Contribution:** 3
**Rating:** 6
**Confidence:** 3

**Summary:**

The paper introduces PASTA, a perceptual assessment system designed to benchmark explainable AI (XAI) techniques in a human-centric manner. The authors first integrate annotated images from the COCO, Pascal Parts, Cats Dogs Cars, and Monumai datasets to create a benchmark dataset for XAI, which is also used to train classifier models for generating explanations. For the final assessment of XAI methods, the authors curate a PASTA evaluation benchmark comprising 100 images. They apply 46 distinct combinations of 21 different XAI techniques on these 100 images, creating a dataset of 4,600 instances. Each instance includes an image, its ground truth label, a classifier’s predicted label, and the explanation generated by a specific XAI technique.

Additionally, to compare various XAI methods, including saliency-based and explanation-based approaches, the authors develop an automated evaluation metric that models human preferences based on a database of human evaluations. This metric enables robust evaluations across modalities. The experiments, conducted on 21 different XAI techniques and datasets, demonstrate that saliency-based explanation techniques such as LIME and SHAP align more closely with human intuition than explanation-based methods.

**Strengths:**

1. The paper addresses an important problem of benchmarking XAI methods in a comprehensive and efficient way, aligning evaluations with human assessments, which I believe is an important gap in the literature.

2. The proposed benchmark allows for the comparison of XAI methods across different modalities, facilitating evaluations of both explanation-based and saliency-based methods.

3. The paper introduce a data-driven metric to mimic human assessments in evaluating the interpretability of explanations.

4. The authors created six carefully designed questions and a comprehensive set of criteria to assess various desired properties while collecting human labels.

5. Different XAI methods are evaluated against human assessments to benchmark the quality of their explanations. Additionally, human scores are compared with different XAI metrics.

6. The authors have indicated their willingness to open-source their code, annotations, and models.

**Weaknesses:**

1. How many human subjects were involved in the benchmark creation? Is it five? This information is not explicitly stated in the main paper. Since the benchmark aims to align with human evaluations, it would be valuable to provide details about the annotators, including their ethnicity, gender, age, and other relevant demographics. Ideally, involving annotators from diverse backgrounds—such as varying ages, genders, and ethnicities—would help reduce potential biases in the benchmark.

2. The evaluation benchmark includes only 100 images, which may not be statistically sufficient for conclusive insights into model behavior across different XAI techniques.

3. There appears to be a possibility that some images could overlap between training and test sets. While the authors demonstrated generalization by separating these sets explicitly in Section 4.4, it is not clear to me  why they did not consistently apply such strict separation across all experiments.

4. The method utilizes existing object grounding techniques, such as Grounding DINO, to generate bounding boxes for the “Cats Dogs Cars” dataset, which might lead to suboptimal performance. It is also unclear if the authors conducted any ablation studies on different grounding or object detection methods before selecting Grounding DINO. Was there a particular reason for choosing this specific method?

5. Many essential information related to the proposed approach are not included in the main paper. For example, the complete list of perturbations need to be included in the main paper, not on the appendix.

**Questions:**

1. On page 18, in Appendix A1.1, it’s stated that "the task we focus on is indoor scene labeling," but this was not mentioned in the main paper. Could you please clarify this?

2. The authors mention a low correlation between widely used XAI methods and human assessments. Does this imply that existing metrics do not accurately align with human evaluations? If so, how do the authors conclude that human and metric-based assessments cover complementary aspects? What complementary information is provided by the existing metric-based approaches that human-subject-based evaluation does not capture?

Also, See the weaknesses mentioned above.

---

> ### Author Response · Authors · 2024-11-18
> **Answer to reviewer cj8g**
>
> We would like to thank reviewer cj8g for their thoughtful comments and suggestions which helped us improve the clarity and thoroughness of the paper. Below, we give an answer to each indicated weakness and question.
>
> ---
>
> ### 1. Annotator Demographics and Diversity (W1)
>
> The creation of the benchmark has involved 15 human subjects, each of whom has annotated a part of the whole dataset (roughly one third). For each sample of the dataset, 5 different annotators among the 15 have made an annotation. The precise demographics of the annotators are described in Appendix A.3. In particular, the genders of the annotators were willingly chosen to be balanced.
>
> | Age Range   | Number of Participants |
> |-------------|-------------------------|
> | 0-22    	| 3                   	|
> | 22-25   	| 5                   	|
> | 25-30   	| 4                   	|
> | 30-100  	| 3                   	|
>
> | Sex	| Number of Participants |
> |--------|-------------------------|
> | male   | 5                   	|
> | female | 10                  	|
>
> ---
>
> ### 2. Sample Size of the Benchmark (W2)
>
> We would like to clarify an important point. While traditional datasets are often evaluated based solely on the number of images, our benchmark incorporates two additional dimensions: the number of annotators and the number of XAI techniques. Moreover, we account for the number of questions, which directly impacts the total number of annotations per data point. Consequently, although our benchmark includes only 100 images sourced from 4 different datasets, it features 5 independent answers to 6 distinct questions across 22 XAI techniques. This results in a total of 66,000 human annotations. This annotation scale is comparable to that of MMbench [1], which involves approximately 3,000 images.
>
> Rather than increasing the number of images, we prioritized a comprehensive evaluation of XAI techniques and coverage of 6 questions addressing different XAI axioms—an original contribution to the literature. To provide a clearer comparison with existing benchmarks, we have included a detailed analysis in the appendix (Section: Comparison with Existing Benchmarks). This analysis lists various XAI benchmarks [4-11] present in the literature. The following table demonstrates that our benchmark is comparable to these existing benchmarks:
>
> | Name               	| Annotations 	| N_Samples | N_Part | Modality | N_Q | N_XAI | N_Data |
> |------------------------|-----------------|-----------|--------|----------|------|-------|--------|
> | PASTA-dataset      	| Likert      	| 66,000	| 15 	| I + C	| 6	| 21	| 100	|
> | Yang et al. (2022) [4] | Saliency, 2AFC  | 356   	| 46 	| I    	| 2	| 1 	| 89 	|
> | Colin et al. (2022) [5]| Classification  | 1,960 	| 241	| I    	| 1	| 6 	| NA 	|
> | Dawoud et al. (2023) [6]| Clicktionary   | 3,836 	| 76 	| I    	| 1	| 3 	| 102	|
> | Mohseni et al. (2021) [7]| Saliency  	| 1,500 	| 200	| I + T	| 1	| No	| 1,500  |
> | Herm et al. (2021) [8] | Likert      	| NA    	| 165	| C    	| 1	| 6 	| NA 	|
> | Morrison et al. (2023) [9]| Click/QCM	| 450   	| 50 	| I    	| 1	| 3 	| 39 	|
> | Spreitzer et al. (2022) [10]| Likert/Binary| 4,050	| 135	| C    	| 9	| 2 	| NA 	|
> | Xuan et al. (2023) [11]| Likert/Binary   | 3,600 	| 200	| C    	| 4	| 2 	| 1,326  |
>
> Regarding the types of human annotations used in existing studies (as shown in the "Annotations" column of the table), Likert refers to the use of a Likert scale for scoring, Saliency refers to pseudo-saliency map evaluations, 2AFC indicates two-alternative forced choices, Clicktionary corresponds to the click-based annotation game defined in [4], MCQ represents multiple-choice questions, and Binary refers to binary decision tasks. Our novelty lies in the larger volume of annotations compared to previous approaches, the large number of XAI techniques, and the use of transparent, post-hoc methods.
>
> ---
>
> ### 3. Data Splitting and Generalization (W3)
>
> As explained in Sec. 4.4, in the experiment, if no split restriction is given, it is possible that the same images with different XAI techniques, or the same XAI techniques applied to different images could be found in the training and test set. As can be seen in Table 4, adding a split restriction to avoid this has a small, but non-zero effect on the accuracy of the metric. To simplify this, we have chosen in the updated version to remove the ‘’no restriction” and only consider the restriction about images in all our experiments. As such, all the values in Section 4.3 and Appendix C have been replaced. The observations remain the same.
>
> ---
>
> ### 4. Choice of Grounding DINO Technique (W4)
>
> To generate bounding boxes, among the different available solutions, Grounding DINO is one of the best methods [2]. In addition, since the Cats Dogs Cars dataset is of moderate size, we have manually checked the correctness of the bounding boxes produced by Grounding DINO and found no significant errors.

---

> > ### Author Response · Authors · 2024-11-18
> > **Second part of the answer to reviewer cj8g**
> >
> > ### 5. Completeness of Information in the Main Paper (W5)
> >
> > The complete list of perturbations used to generate the dataset was indeed missing from the manuscript. We thank the reviewer for bringing this to our attention. We have added it in the updated version in Appendix A.4.
> >
> > ---
> >
> > ### 6. Clarification of Task and Focus (Q1)
> >
> > The sentence that you mentioned was only related to the COCO dataset and not all of them, but it was ambiguous. We have corrected this in the updated version.
> >
> > ---
> >
> > ### 7. Alignment Between XAI Metrics and Human Assessments (Q2)
> >
> > We observed a low correlation between the existing metrics we tested (namely ROAD [12], MaxSensitivity [13] and Sparseness [14]) and human evaluations, which aligns with prior findings reported by [3]. We believe this discrepancy arises because these metrics measure complementary aspects. Human evaluations are most suited to assess the usefulness of an explanation, given their focus on human interpretability. In contrast, computational metrics like faithfulness primarily evaluate how well the explanation aligns with the actual functioning of the model. This indicates that human perception of image perturbations relies on visual attributes different from those used by current XAI evaluation techniques. While XAI metrics focus on the model's inner workings, human perception is more likely influenced by how the image is interpreted. Additionally, human annotations are often context-dependent (e.g., varying with the type of images), leading to variability across stimuli that may not affect XAI metrics as much. We have included a clarification on this point in Section 3.6.
> >
> > ---
> >
> > ### References
> >
> > [1] Liu, Y., Duan, H., Zhang, Y., Li, B., Zhang, S., Zhao, W., ... & Lin, D. (2025). Mmbench: Is your multi-modal model an all-around player?. In European Conference on Computer Vision (pp. 216-233). Springer, Cham.
> > [2] https://paperswithcode.com/sota/zero-shot-object-detection-on-mscoco
> > [3] Felix Biessmann and Dionysius Refiano. Quality metrics for transparent machine learning with and without humans in the loop are not correlated. arXiv preprint arXiv:2107.02033, 2021.
> > [4] Scott Cheng-Hsin Yang, Nils Erik Tomas Folke, and Patrick Shafto. A psychological theory of explainability. In International Conference on Machine Learning, pp. 25007–25021. PMLR, 2022.
> > [5] Julien Colin, Thomas Fel, R´emi Cad`ene, and Thomas Serre. What I cannot predict, I do not understand: A human-centered evaluation framework for explainability methods. Advances in NeuralInformation Processing Systems, 35:2832–2845, 2022.
> > [6] Karam Dawoud, Wojciech Samek, Peter Eisert, Sebastian Lapuschkin, and Sebastian Bosse. Human-centered evaluation of XAI methods. In 2023 IEEE International Conference on Data Mining Workshops (ICDMW), pp. 912–921. IEEE, 2023.
> > [7] Sina Mohseni, Jeremy E. Block, and Eric Ragan. Quantitative evaluation of machine learning explanations: A human-grounded benchmark. In 26th International Conference on Intelligent User Interfaces, pp. 22–31, 2021.
> > [8] Lukas-Valentin Herm, Jonas Wanner, Franz Seubert, and Christian Janiesch. I don’t get it, but it seems valid! the connection between explainability and comprehensibility in (x) ai research. In ECIS, 2021
> > [9] Katelyn Morrison, Mayank Jain, Jessica Hammer, and Adam Perer. Eye into ai: Evaluating the interpretability of explainable ai techniques through a game with a purpose. Proceedings of the ACM on Human-Computer Interaction, 7(CSCW2):1–22, 2023.
> > [10] Nina Spreitzer, Hinda Haned, and Ilse van der Linden. Evaluating the practicality of counterfactual explanations. In XAI. it@ AI* IA, pp. 31–50, 2022.
> > [11] Yueqing Xuan, Edward Small, Kacper Sokol, Danula Hettiachchi, and Mark Sanderson. Can users correctly interpret machine learning explanations and simultaneously identify their limitations? arXiv preprint arXiv:2309.08438, 2023
> > [12] Yao Rong, Tobias Leemann, Vadim Borisov, Gjergji Kasneci, and Enkelejda Kasneci. A consistent
> > and efficient evaluation strategy for attribution methods. arXiv preprint arXiv:2202.00449, 2022.
> > [13] Yeh, C. K., Hsieh, C. Y., Suggala, A., Inouye, D. I., & Ravikumar, P. K. (2019). On the (in) fidelity and sensitivity of explanations. Advances in neural information processing systems, 32.
> > [14] Chalasani, P., Chen, J., Chowdhury, A. R., Jha, S., & Wu, X. (2018). Concise explanations of neural networks using adversarial training. arXiv arXiv–1810.

---

> > > ### Author Response · Authors · 2024-11-23
> > >
> > > Dear Reviewer cj8g,
> > >
> > > Thank you for taking the time to serve as a reviewer for our paper. We would like to kindly remind you that the rebuttal period will conclude in less than a week. As of now, we have not received any feedback from you. Could you please share your comments or suggestions with us?
> > >
> > > Best regards,

---

> ### Author Response · Authors · 2024-12-02
> **A kindly reminder**
>
> Dear reviewer cj8g,
>
> Thank you for serving as a reviewer for our paper. We wanted to kindly remind you that it has been two weeks since we submitted our response and we have not yet received any feedback.
>
> We value your insights and suggestions. If there are any additional points you would like us to clarify or discuss, please do not hesitate to let us know.
>
> Best regards,

---

### Author Response · Authors · 2024-11-18
**General Answer**

Dear AC and reviewers,

We would like to express our sincere thanks for your constructive comments and questions regarding our work. We greatly appreciate the reviewers’ feedback and are grateful for the opportunity to address the concerns raised.

We believe that PASTA and the dataset could serve as a valuable benchmark for the community to assess the quality of XAI techniques. In response to the specific questions raised by the reviewers, we have made the following updates and clarifications (in purple in the revised manuscript):

- **In the appendix (Section B.1 Comparison with Existing Benchmarks):** We have added a comparison between the PASTA dataset and existing human-aligned evaluation benchmarks.
- **In Section A.4 (PERTURBATIONS) of the appendix:** We have included the list of perturbations and additional details on their usage.
- **In Section 3.6 (CORRELATION WITH OTHER METRICS):** We have clarified that human evaluations and computational metrics should be viewed as complementary aspects of the evaluation process.
- **In Section 4.3 (CLASSIFIER RESULTS):** We have tested variants of the feature extraction process, including replacing CLIP with the LLaVa encoder and using handcrafted feature extraction. Details on how we compute these handcrafted features are provided in Section D (VARIANT WITH HANDCRAFTED FEATURES).
- **In Section C.4 (SCORING FUNCTIONS):** We tested alternative scoring networks, including Ridge Regression, Lasso Regression, Support Vector Machines, and a Multi-Layer Perceptron with a single hidden layer of 100 units.
- **In Section B.2 (COMPARISON WITH EXISTING METRICS):** We have added additional formulas and figures to better illustrate the process of computing our scores.
- **In Section B.5 (EVALUATION QUESTIONS):** We provide insights into annotators' feedback and their opinions on XAI, as well as the questions they were asked to answer.
- **In Section A.1.1 (DATASET):** We have clarified the classes and concepts used in the datasets for training our inference models, and we have included a table of labels.
- **In Section C.3:** We have added an experiment measuring the impact of label information on the embedding in the PASTA metric.
- **We have expanded our analyses** in Sections 3.5 and 3.6.
- **In the conclusion:** We have highlighted interesting avenues for future work, including the potential for PASTA-metric as a perceptual loss.
- **In Section 3.2 (XAI METHODS):** We provide further clarification on our testing across multiple backbones.
- **We have re-run all our experiments** related to the PASTA-metric, considering the restrictions on images as indicated by $img_{id}$ in Table 4. As such, all values have been updated, though the observations remain consistent.
- **We have corrected some typos.**
- **In Section B.2 (COMPARISON WITH EXISTING METRICS):** We have added more detailed information on how the computational metrics were obtained.

By Wednesday, we plan to make the following revisions to the paper:
- Correct the typo in Figure 8.
- Include examples of both high- and low-rated explanations, showcasing human annotations alongside PASTA-metric ratings.
- Add more detailed examples in the main text before referencing supplementary sections.

We sincerely hope the reviewers will appreciate the efforts we've made to address their feedback. Once again, we deeply appreciate the reviewers’ thorough feedback. We look forward to engaging in further discussions with the reviewers. We are happy to address any further questions and look forward to your continued input.

Best regards,

---

> ### Author Response · Authors · 2024-11-20
> **Update**
>
> Dear AC and Reviewers,
>
> Following the plan outlined in the previous post, we have implemented the following changes:
>
> - Added examples of both high- and low-rated explanations, showcasing human annotations alongside PASTA-metric ratings (Appendix E).
> - Included more detailed examples in the main text to provide better context before referencing supplementary sections.
> - Corrected typos throughout the manuscript.
> - The typo in Figure 8 will be corrected tomorrow.
>
> Thank you for your feedback and guidance.

---

> ### Author Response · Authors · 2024-11-21
> **A kindly reminder**
>
> Dear Reviewers,
>
> We hope this message finds you well.
>
> We have revised our manuscript to address the concerns and suggestions you raised. We greatly value your expertise and would highly appreciate it if you could review the updated manuscript and share your thoughts on whether these revisions satisfactorily address your concerns.
> We sincerely thank the reviewers for their efforts and would greatly appreciate it if they could begin engaging in a discussion with us.
> Please feel free to reach out if you have any further questions or require additional clarification.
>
> Best regards,
>
> The corresponding authors

---

> ### Author Response · Authors · 2024-11-26
> **Additional update**
>
> We sincerely thank reviewers cj8g, 1rMP, cta3, and 2wcf for their insightful comments and suggestions. In addition to the revisions detailed in our previous responses (and based on your feedback), we have conducted new experiments to investigate potential applications of the PASTA-score, inspired by the reviewers' suggestions. Specifically, we added the following experiments in **Section E.2**:
> - **Experiment 1**: Using the PASTA-score to automate the selection of the optimal kernel width for the exponential kernel used in blurring images during the LIME perturbation set optimization process.
> - **Experiment 2**: Identifying among GradCAM explanations on different layers of ResNet-50, the one with the highest PASTA-score and evaluating whether the resulting explanations align with expectations.
>
> Additionally, we have expanded the conclusion to outline further avenues for future research.
>
> Once again, we sincerely thank you for your constructive feedback and would be happy to continue the discussion with you.
>
> best regards

---

> ### Author Response · Authors · 2024-11-30
> **Summary of changes in the revised manuscript**
>
> As the rebuttal period comes to a close, we would like to provide a comprehensive summary of the changes made to our manuscript based on the insightful feedback from the reviewers. We remain open to further discussions should there be any additional issues.
>
> Changes in the revised manuscript are marked in **purple** for clarity.
>
> ## **Precision About Contributions to the State of the Art**
>
> One key concern was clarifying the contribution of the PASTA-dataset to the field of perceptual evaluation of XAI methods. To address this:
> We added in **Section B.1** a detailed comparison between the PASTA-dataset and existing human-aligned evaluation benchmarks.  The PASTA-dataset is, to our knowledge, the first dataset dedicated to perceptual evaluation of XAI methods that integrates both saliency and concept-based explanations.  It combines **2,200 samples from 21 XAI methods** and incorporates **6 questions carefully crafted with input from a psychologist** to assess perceptual quality.  We also emphasize the importance of collecting **5 annotations per sample**, resulting in **66,000 Likert annotations**, far exceeding the scale of existing works.  The choice of multiple annotations per sample, despite being costly, seems relevant considering the high variance inherent to perceptual assessment, as underlined in Section B.3.
>
> ## **Clarity Issues**
>
> Several points in the original manuscript were unclear, and we made the following improvements:
>
>    - **Section 3.6**: We clarified our interpretation of the lack of correlation between PASTA-dataset annotations and existing computational metrics, we have placed more emphasis on the fact that we consider that human evaluations and computational metrics should be viewed as complementary aspects of the evaluation process.
>
>    - **Sections 3.5 and 3.6**: We expanded the analysis of results to address the reviewers’ request for a deeper interpretation.
>
>    - **Section 3.2**:  Clarified why and how we tested XAI methods across multiple backbones.
>
>    - Corrected typos.
>
>    - Added more detailed examples in the main text for better context before referencing supplementary sections.
>
>    - **Section B.2**: Additional formulas and figures were included to illustrate the process of computing the PASTA-scores more clearly.
>
> ## **Details About the PASTA-Dataset Process**
>
> Thanks to reviewers suggestions, we provided more detailed information about the dataset creation:
>
>    - **Section A.1.1**: Clarified the classes and concepts used for training inference models and included a table of labels. Note that the same images and labels are used across all models.
>    - **Section A.4** In response to a reviewer’s observation about the importance of perturbations in Q5 and Q6, we added a detailed list of perturbations and their usage.
>
> ## **Testing Variants of the PASTA-Metric**
>
> To address concerns about the robustness of the PASTA-metric, we conducted additional experiments:
>
> 1. **Bias From CLIP Encoding**:  In section **Section 4.3**, we added experiments to investigate the potential bias introduced by using CLIP as a multimodal encoder.
>
>   	- Study the replacement of CLIP with another multimodal encoder, LLaVa; results were similar.
>   	- Proposed an alternative using handcrafted features in **Section D**, addressing concerns about potential biases of DNN-based encoders.
>
> 2. **Scoring Network Alternatives**: Also in Section D, we tested Ridge Regression, Lasso Regression, Support Vector Machines, and a Multi-Layer Perceptron.  Then, we demonstrated that our proposed network, with ranking and cosine similarity losses, performed better.
>
> 3. **Impact of Label Information**: In **Section C.3**: We tested the effect of adding label information to the PASTA-metric network. Results showed no improvement, likely due to the large number of classes.
>
> 4. **Bias in Data Splits**:  **For all of our experiments**, we Re-ran computations to ensure no overlap between samples from the same image but different XAI methods in training, validation, and test splits. Updated values are consistent with previous observations.

---

> > ### Author Response · Authors · 2024-11-30
> > **Summary of changes in the revised manuscript (second part)**
> >
> > ## **Additional Study About User Feedback**
> >
> > To address concerns about annotator profiles, we provided details about annotators' feedback and their opinions on XAI, along with the questions they answered in **Section B.5**. We hope that transparency helps the future reader to perceive the annotator's mindset.
> >
> > ## **Proposed Applications of the PASTA-Metric**
> >
> > In response to requests for more practical applications of the PASTA-metric, we first Highlighted in the conclusion as a promising direction for future work the use of the PASTA-metric as a perceptual Loss for XAI.  Regarding uses cases, in  **Appendix E**, we first evaluated the PASTA-metric on unseen images alongside corresponding PASTA-metric ratings, demonstrating the method's viability. Then we proposed two additional experiments:
> >
> > - Use the PASTA-score to optimize the kernel width for the LIME perturbation process.
> > - Identify the ResNet-50 layer with the highest PASTA-score and evaluate whether the explanations align with expectations.
> >
> > In both cases, visual results corroborate the improvements suggested by the PASTA-score.
> >
> > We hope these changes address your feedback comprehensively and are sincerely happy looking forward to addressing any constructive discussion if there remains some points that do not convince you.

---

### Meta-Review · Area_Chair_EDMs · 2024-12-18

**Metareview:**

The submission initially had mixed reviews, and the major concerns raised are:

1. limited benchmark size (5 humans, 100 images) [cj8g, 2wcf]
2. share images in training/test sets? [cj8g]
3. missing ablation study on grounding method [cj8g]
4. missing explanation/analysis about why there is low correlation between XAI methods and human assessments. [cj8g, 2wcf]
5. need to use other gradient-based method for transformers [cta3]
6. limited usefulness since new human annotations would be needed for new XAI methods [cta3]
7. uses CLIP to evaluate explanations, which limits the model to the limitations/biases of CLIP (or whatever VLM is used) [cj8g, 2wcf]
8. perhaps more suitable for user-based study conference rather than ICLR [cj8g]
9. how to use the method to evaluate XAI approaches in different domains? limited generalization? [2wcf, 1rMP]
10. shallow analysis, lacks insight [2wcf]
11. missing comparison with other regression methods [2wcf, 1rMP]

The authors wrote a response, and after the discussion period, the reviewers were still split on the paper. One major concern (Point 9) raised by a majority of reviewers was about the generalizability of the PASTA-metric to other XAI approaches and other domains. Another concern (Point 7) was the use of CLIP as a feature extractor for regressing the evaluation scores for other XAI approaches, and the potential bias introduced by this. (Although authors provide results with LLAVA, it should be noted that LLAVA is based on CLIP). Finally, the suitability of the user-study-based work (Point 8) was also mentioned.

Regarding Point 7, the AC also agrees that using CLIP or other VLMs as feature extractors makes the generated metric for new XAI methods potentially biased towards high-level semantic content (and the limitations of CLIP), especially as the image contents are visible in the input heat map. It is unclear why the training data couldn't be used to train / fine-tune a basic CNN regressor.

Regarding Point 8, the paper consists of a user-study of XAI methods (PASTA dataset), and a method to predict the user evaluations on new XAI methods (PASTA metric). In this sense, the paper seems to be more suitable for a user-study based conference -- the main goal is to evaluate XAI in terms of user preferences. The authors argue that "the objective is not to interpret human annotations but rather to assess the performance of XAI techniques."  However, this is not entirely convincing since only the human-interpretability performance is evaluated here, whereas the primary focus of XAI methods should be faithfulness to the explained model.

Overall, the paper still has a few outstanding problems that need to be carefully addressed.

As a side note, there is additional related work that evaluates plausibility of XAI methods using human eye gaze:
https://ojs.aaai.org/index.php/HCOMP/article/view/22002

**Additional Comments On Reviewer Discussion:**

See above

---

### Decision · Program_Chairs · 2025-01-22

Reject